# ZFP281-BRCA2 prevents R-loop accumulation during DNA replication

Yan Wang[1], Binbin Ma[2], Xiaoxu Liu[1], Ge Gao[3], Zhuanzhuan Che[1], Menghan Fan[1], Siyan Meng[1], Xiru Zhao[1], Rio Sugimura [3], Hua Cao[4], Zhongjun Zhou [3], Jing Xie[2], Chengqi Lin [4,5,6✉] & Zhuojuan Luo [1,6✉]

R-loops are prevalent in mammalian genomes and involved in many fundamental cellular processes. Depletion of BRCA2 leads to aberrant R-loop accumulation, contributing to genome instability. Here, we show that ZFP281 cooperates with BRCA2 in preventing R-loop accumulation to facilitate DNA replication in embryonic stem cells. ZFP281 depletion reduces PCNA levels on chromatin and impairs DNA replication. Mechanistically, we demonstrate that ZFP281 can interact with BRCA2, and that BRCA2 is enriched at G/C-rich promoters and requires both ZFP281 and PRC2 for its proper recruitment to the bivalent chromatin at the genome-wide scale. Furthermore, depletion of ZFP281 or BRCA2 leads to accumulation of R-loops over the bivalent regions, and compromises activation of the developmental genes by retinoic acid during stem cell differentiation. In summary, our results reveal that ZFP281 recruits BRCA2 to the bivalent chromatin regions to ensure proper progression of DNA replication through preventing persistent R-loops.

[1] Jiangsu Provincial Key Laboratory of Critical Care Medicine, Key Laboratory of Developmental Genes and Human Disease, School of Life Science and Technology, Southeast University, Nanjing 210096, China. [2] Institute for Regenerative Medicine, Shanghai East Hospital, Frontier Science Center for Stem Cell Research, School of Life Sciences and Technology, Tongji University, Shanghai 200092, China. [3] School of Biomedical Sciences, LKS Faculty of Medicine, The University of Hong Kong, 21 Sassoon Road, Hong Kong 999077, China. [4] Key Laboratory of Technical Evaluation of Fertility Regulation of Non-human primate, Fujian Provincial Maternity and Children's Hospital, Affiliated Hospital of Fujian Medical University, Fuzhou 350005, China. [5] Jiangsu Province Hi-Tech Key Laboratory for Biomedical Research, Key Laboratory of Developmental Genes and Human Disease, School of Life Science and Technology, Southeast University, Nanjing 210096, China. [6] Shenzhen Research Institute, Southeast University, 19 Gaoxin South 4th Road, Nanshan District, Shenzhen 518063, China. ✉email: cqlin@seu.edu.cn; zjluo@seu.edu.cn

R-loop is a three-stranded nucleic acid structure formed accompanying transcription by threading back of nascent RNA into the template DNA, with the non-template DNA unpaired and displaced[1]. Genome-wide R-loop profiling has demonstrated that R-loops are prevalent and conserved in mammalian genomes[2–4]. R-loop formation and accumulation are the outcome of intricate interplay of transcription process, DNA sequence, and chromatin structure[5,6]. For years, R-loops were thought to be natural and trivial byproducts of transcription. In the past decade increasing evidence indicates that R-loop formation leads to DNA replication stalling, and thus detrimental to genome stability[4,7,8]. Aberrant R-loop accumulation has been linked to increased DNA breaks, transcription-replication collisions, mutagenesis, and chromosome rearrangement[9–11].

Many factors have been identified in preventing excessive R-loops, including RNase H enzymes, DNA:RNA helicases, RNA binding and processing proteins, and factors in the genome protection pathway[12–14]. For example, members in the Fanconi Anemia pathway, such as BRCA2, have emerged as important R-loop regulators[15,16]. *BRCA2* germline mutations are known to predispose to high risk of breast, ovarian and other cancers[17,18]. BRCA2-deficient cells show severe chromosome abnormalities, including chromosome breaks, centrosome amplification and aneuploidy[19,20]. It has been demonstrated that BRCA2 functions as a tumor suppressor in maintaining genome integrity, through homologous DNA repair and replication fork protection[21,22]. Recent studies also indicated that loss of BRCA2 leads to unscheduled accumulation of R-loops, proposing R-loops as a trigger of chromosome abnormalities in BRCA2-deficient cells[14]. Although extensive efforts have been invested in the functions and the regulations of R-loops, it remains largely unclear whether a specific regulatory mechanism exists under distinct chromatin states, and how it is linked to the diverse physiological and pathological function of R-loops.

The krüppel-like zinc finger transcription factor ZFP281 was previously identified as a transcriptional regulator involved in multiple cellular processes[23–27]. ZFP281 regulates stem cell state transitions through c-MYC, TET1/2 or EHMT1-ZIC2[28,29]. ZFP281 also induces epithelial-mesenchymal transition (EMT) in colorectal cancer cells via activating the EMT related genes[30]. In addition, ZFP281 is directly or indirectly involved in DNA damage response via XRCC4[31,32]. Mechanically, our previous studies have indicated that ZFP281 is able to recruit c-MYC and the Super Elongation Complex (SEC) to chromatin to regulate transcriptional elongation[25,26]. ZFP281 can also suppress repeat element LINE-1 expression via TET1/MLL2 in embryonic stem (ES) cells[27].

Here we found that ZFP281 knockout leads to DNA replication defects and early S phase delay, and impairs cell proliferation. Further analyses suggested that R-loop accumulation might account for DNA replication defects in ZFP281 knockout cells. To investigate the mechanism by which ZFP281 prevents excessive R-loop, we identified the ZFP281 interacting proteins by affinity purification and mass spectrometry, and revealed that ZFP281 and BRCA2 are able to interact with each other. ZFP281 recruits BRCA2 to the bivalent chromatin to prevent unscheduled R-loop formation and ensure progression of DNA replication. In addition, depletion of either ZFP281 or BRCA2 compromises the activation of bivalent genes during stem cell differentiation. Our data indicated a specific role of ZFP281 in recruiting BRCA2 to suppress R-loop formation at the bivalent chromatin and maintain proper DNA replication.

## Results

**ZFP281 depletion leads to early S phase delay.** Two independent lines of ZFP281 knockout (KO) cells were generated using the CRISPR/Cas9 system to study the function of ZFP281 in mouse ES cells (Fig. 1a and Supplementary Fig. 1a, b). We noticed that ZFP281 KO cells tended to grow slower than wild type (WT) cells (Supplementary Fig. 1c). Further cell proliferation analyses indicated that ZFP281 KO cells showed a significant reduction in total cell numbers after culture for 72 h, despite starting from the same number of cells as WT cells (Fig. 1b).

We next examined the effects of ZFP281 depletion on cell cycle progression by flow cytometry analyses, which showed a slight reduction in 5-ethynyl-2'-deoxyuridine (EdU) labeled S phase cell fraction from 68% to about 61% after ZFP281 KO (Fig. 1c and Supplementary Fig. 1d). We also examined the expression of the S phase specific Cyclin E1 and E2 after ZFP281 KO. Paralleling its RNA levels, Cyclin E1 protein levels were reduced in the two ZFP281 KO cells, though the RNA levels of Cyclin E2 remained unchanged (Fig. 1d and Supplementary Fig. 2). The protein levels of the S phase specific Cyclin A2 were also reduced after ZFP281 KO, while the G1 phase specific Cyclin C was marginally affected (Fig. 1d and Supplementary Fig. 2). To further investigate whether ZFP281 is required for proper S phase progression during cell cycle, we identified early, middle and late S phase cells according to the spatiotemporal pattern of EdU incorporation, which has been well-defined for indicating the DNA replication timing[33,34]. Compared with WT ES cells, ZFP281 KO cells exhibited a significant increase in the percentage of early S phase cells, in which EdU labeled replication foci are small and spreading throughout the nucleus (Fig. 1e). In contrast, the proportion of middle S phase cells with larger but less replication foci was decreased in ZFP281 KO cells (Fig. 1e). However, the EdU positive ratios of entire S phase were even slightly reduced after ZFP281 KO (Fig. 1c and e). Therefore, we hypothesized that ZFP281 might be required for DNA replication and proper early S phase progression in mouse ES cells.

**ZFP281 depletion leads to DNA replication defects.** We next asked whether ZFP281 depletion delays early S phase progression through affecting DNA replication. Co-immunofluorescent staining of ZFP281 and proliferating cell nuclear antigen (PCNA), which is an essential DNA replication accessory and processivity factor[35], was performed in mouse ES cells. Airyscan superresolution imaging of S phase nucleus showed that ZFP281 was largely co-localized with the PCNA foci (Fig. 2a, b). To examine the function of ZFP281 on DNA replication during early S phase, mouse ES cells were treated with the DNA polymerase inhibitor aphidicolin (APH) to inhibit DNA replication, then released from the replication block by culturing in APH-free media for different time periods. Western analysis showed that the level of chromatin-bound ZFP281 was dropped off after APH treatment, but rapidly restored once re-entry into S phase (Fig. 2c), further supporting the enrichment of ZFP281 at replicating chromatin. Interestingly, ZFP281 depletion led to displacement of PCNA from the chromatin fraction to the soluble fraction, without changing the total PCNA levels in whole cell lysates (Fig. 2d and Supplementary Fig. 3a). In addition, reciprocal immunoprecipitations demonstrated that ZFP281 was able to interact with PCNA (Fig. 2e). We performed the proximity ligation assay (PLA) to probe the direct interaction between ZFP281 and PCNA in mouse ESCs. The ZFP281-PCNA PLA fluorescent puncta were formed and evident in mouse ES cells. APH treatment reduced the number of the ZFP281-PCNA PLA fluorescent puncta (Fig. 2f). Thus, our results suggest that ZFP281 might be directly required for maintain the proper levels of PCNA on chromatin during DNA replication.

To examine whether ZFP281 is able to directly bind to nascent DNAs, we performed re-ChIP assay using 5-bromo-2'-

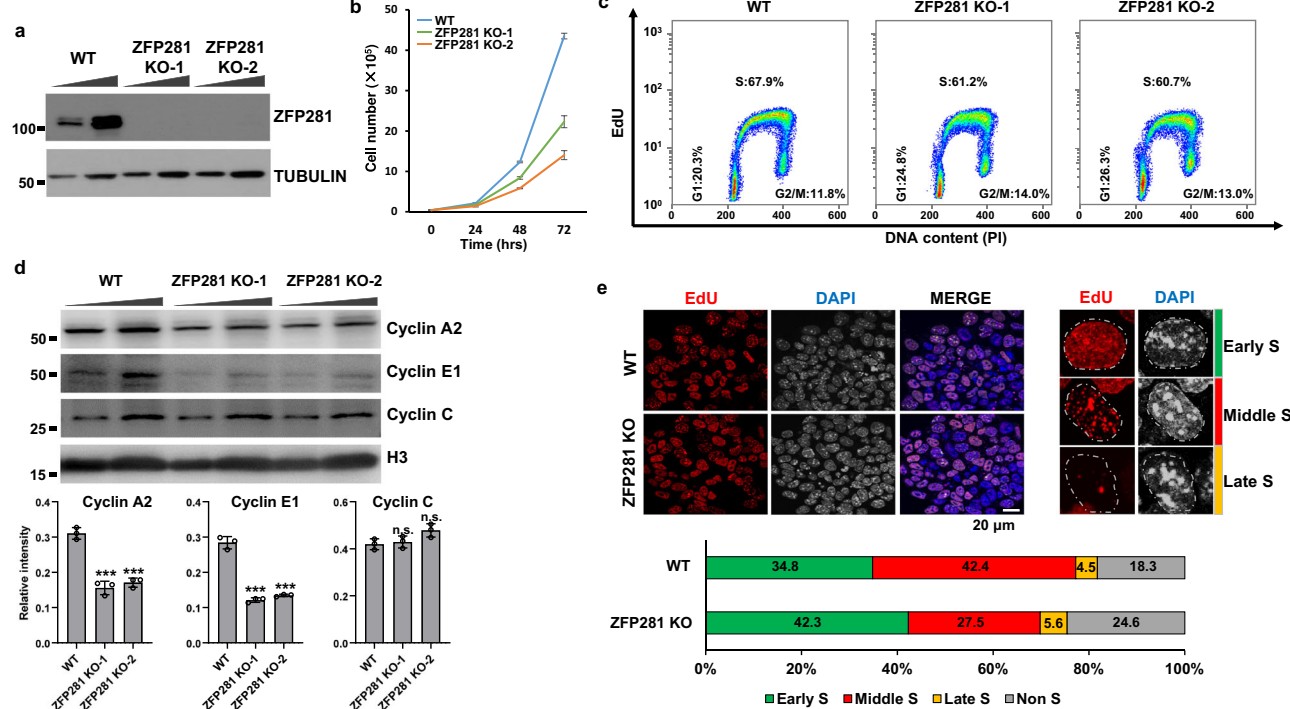

**Fig. 1 ZFP281 is required for proper S phase progression in mouse ES cells. a** Western blot showing successful depletion of ZFP281 in mouse ES cells by CRISPR-Cas9. α-TUBULIN was used as a loading control. Four independent experiments show similar results. **b** Growth curve showing numbers of WT (yellow), ZFP281 KO-1 (green), and ZFP281 KO-2 (red) mouse ES cells counted at different time intervals after plating. Mean ± SEM from three independent experiments. **c** Flow cytometry analysis showing the percentage of WT, ZFP281 KO-1, and ZFP281 KO-2 ES cell at different cell cycle phases. Three independent experiments show similar results. **d** Western blot analysis (upper panel) showing the levels of Cyclin A2, Cyclin E1 and Cyclin C in WT, ZFP281 KO-1, and ZFP281 KO-2 ES cells. Histone H3 was used as a loading control. Gray values (lower panel) of Cyclin A2, Cyclin E1 and Cyclin C protein bands were determined by Image J and normalized to the loading control. Mean ± SEM from three independent experiments. Two-tailed, unpaired Student's $t$ tests were performed. (Cyclin A2, WT vs. ZFP281 KO-1, $p = 0.0005$; WT vs. ZFP281 KO-2, $p = 0.0003$. Cyclin E1, WT vs. ZFP281 KO-1, $p = 0.0001$; WT vs. ZFP281 KO-2, $p = 0.0001$. Cyclin C, WT vs. ZFP281 KO-1, $p = 0.6566$; WT vs. ZFP281 KO-2, $p = 0.0500$.) *$p < 0.05$, **$p < 0.01$, ***$p < 0.001$, n.s. = not significant. **e** EdU staining pattern (upper left) of WT and ZFP281 KO ES cells, DNA was counterstained using DAPI; Representative images showing the EdU staining pattern (upper right) of mouse ES cells at early, middle and late S phase; Percentage of WT (total $n = 422$) and ZFP281 KO ES (total $n = 338$) cells at different stages of S phase. Three independent experiments show similar results. Source data are provided as a Source Data file.

deoxyuridine (BrdU) labeled nascent DNAs, and found that ZFP281 was associated with nascent DNAs in all of the tested regions (Fig. 2g and Supplementary Fig. 3b). Moreover, BrdU IP-qPCR analyses revealed that replication recovery from the APH block was affected in ZFP281 KO cells, as reflected by the reduced levels of nascent DNAs at the ZFP281 bound loci *Gata6*, *Kctd1*, *Lhx4* and *Pou3f2* (Fig. 2h).

**ZFP281 prevents R-loop accumulation.** We and other groups have previously reported that ZFP281 is highly enriched at GC-rich chromatin regions[26,36]. G/C skewed CpG island (CGI) promoters and termination sites are hotspots prone to R-loop formation[2,37]. We thus examined whether the S phase delay in ZFP281 KO cells is a potential readout of R-loop formation caused by ZFP281 deletion. Immunostaining of R-loop with the anti-DNA-RNA hybrid S9.6 antibody confirmed substantial increases in R-loop levels after depletion of ZFP281 (Fig. 3a, b and Supplementary Fig. 4).

Previous studies have shown that phosphorylated Histone H3 Ser10 (H3S10P) is not solely an epigenetic marker of condensed chromatin in mitotic cells, but also tightly linked to R-loop accumulation in interphasic cells[38]. To examine whether H3S10P abundance is altered in ZFP281 KO interphasic cells, we performed double immunofluorescence staining to label H3S10P and phosphorylated Histone H3 Thr3 (H3T3P)

simultaneously. As H3T3P can only be detected in mitotic cells[39], the cells negative for H3T3P were considered in interphase. Our analyses showed that the percentage of H3S10P positive interphasic cells were significantly increased after ZFP281 KO (Fig. 3c, d). This is consistent with the finding about accumulation of R-loop after in ZFP281 KO cells (Fig. 3a, b). Overexpression of RNase H1 to dissolve R-loop was able to efficiently inhibit H3S10P accumulation in ZFP281 KO mouse ES cells (Fig. 3c, d).

It has been well established that excessive R-loop can cause DNA damage[40]. Consistently, we also observed that ZFP281 KO cells, either non-treated or treated with APH, accumulated higher levels of the DNA double strand breaks (DSBs) mark γH2A.X, and that RNase H1 overexpression could abrogate the aberrant increase of γH2A.X in ZFP281 KO cells (Fig. 3e and Supplementary Fig. 5). In addition, RNase H1 overexpression could at least partially rescue the defective loading of PCNA to chromatin caused by ZFP281 knockout (Fig. 3e and Supplementary Fig. 5). Taken together, our results suggested that R-loop accumulation might account for DNA replication defects caused by ZFP281 loss.

**Identification of the interaction between ZFP281 and BRCA2.** To better explore the roles of ZFP281 in DNA replication during the early S phase progression, we sought to identify the ZFP281

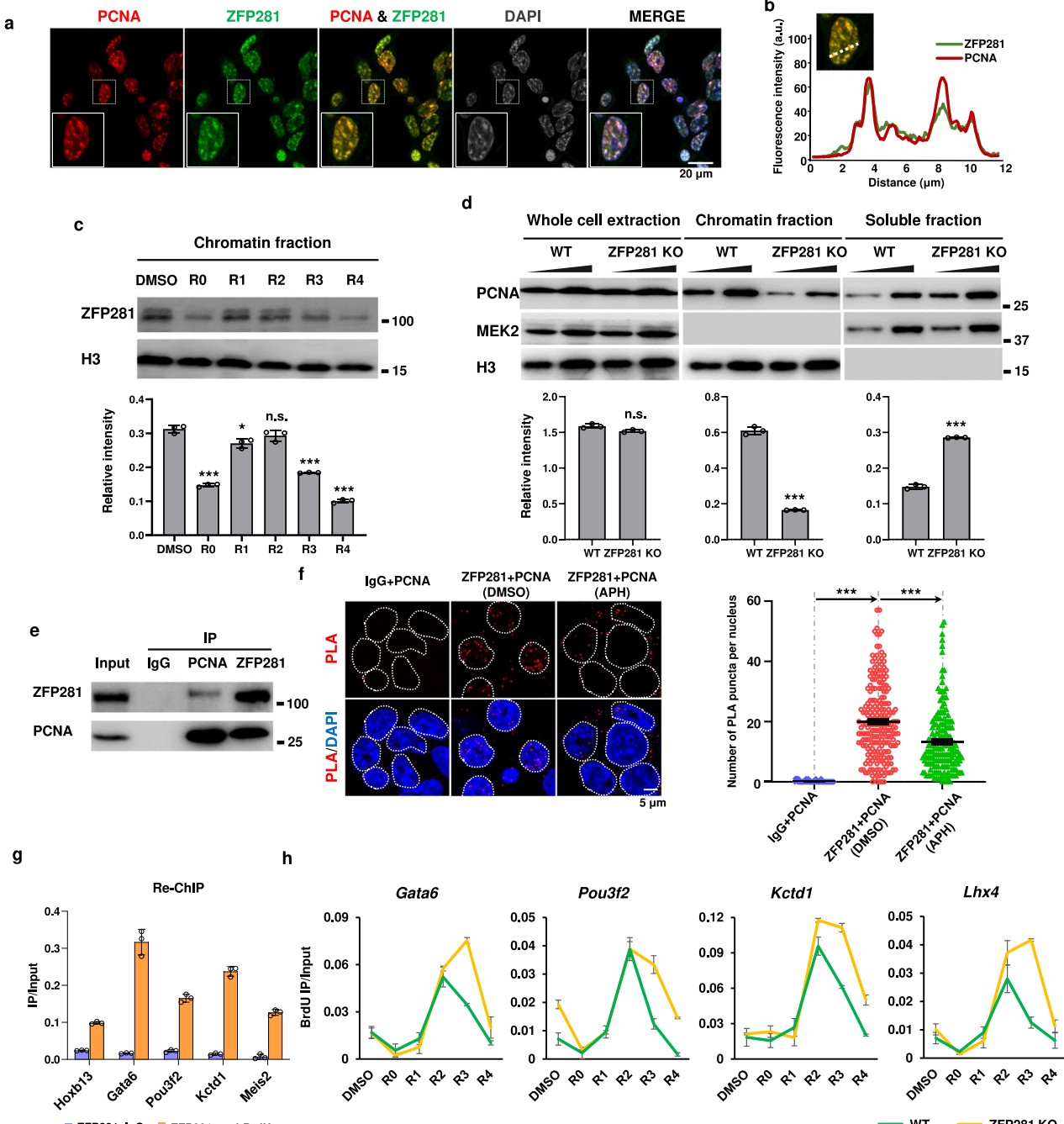

interacting proteins. We generated an inducible stable 293 culture cell line expressing FLAG-tagged ZFP281 using the Flp-In-TRex system[41]. FLAG affinity-purified ZFP281-associated proteins were subjected to silver staining and mass spectrometry analyses, which revealed EMSY, BRCA2, and QSER1 as top candidates for interaction with ZFP281 (Fig. 4a). Consistent with our finding here, the direct interaction between BRCA2 and EMSY has previously been identified by yeast two-hybrid screening[42]. Also, endogenous co-immunoprecipitation from mouse ES cell lysates confirmed the mutual interactions among ZFP281, EMSY, BRCA2, and QSER1 (Supplementary Fig. 6a–d and Fig. 4b). Furthermore, we applied mouse ES cell nuclear extracts to high-resolution size exclusion chromatography, followed by western blot analyses. The results indicated that a portion of ZFP281 was indeed co-eluted with EMSY, BRCA2, and QSER1 from fraction 9 to 12 (> 1.5 MDa) (Fig. 4c).

Subsequently, a series of FLAG-tagged ZFP281 truncated proteins were used to map the domains of ZFP281 interacting with the other three factors (Fig. 4d). The results indicated that both EMSY and BRCA2 were able to interact with the zinc finger domain of ZFP281, while QSER1 required the C-terminus of ZFP281 for the interaction (Fig. 4e and Supplementary Fig. 6e). We also dissected the domains of BRCA2 that interact with ZFP281, QSER1 and EMSY using various BRCA2 truncation mutants, and found that the BRC repeat domain of BRCA2 was essential for their interactions (Fig. 4f, g and Supplementary Fig. 6e). The homologous recombinase RAD51, one of the known interacting partners of BRCA2, binds to the BRC repeat domain in BRCA2 and functions in homologous recombination repair[43–45]. It has been reported that the function of BRCA2 in preventing R-loop accumulation is independent of RAD51[14]. Consistently, only BRCA2, but not RAD51, was detected from the

**Fig. 2 ZFP281 binds to nascent DNAs. a** Co-immunofluorescence showing co-localization of ZFP281 and PCNA in mouse ES cells. DNA was counterstained using DAPI. Three independent experiments show similar results. **b** Profile plot showing normalized pixel intensity of ZFP281 (green) and PCNA (red) corresponding to the region marked with white lines. **c** Western blot analysis (upper panel) showing the level of chromatin-bound ZFP281 in mouse ES cells after release from APH treatment for 0, 1, 2, 3 and 4 hrs respectively. Histone H3 was used as a loading control. Gray values (lower panel) of ZFP281 protein bands were determined by Image J and normalized to the loading control. Mean ± SEM from three independent experiments. Two-tailed, unpaired Student's $t$ tests were performed. (DMSO vs. R0, $p < 0.0001$; DMSO vs. R1, $p = 0.0128$. DMSO vs. R2, $p = 0.1565$. DMSO vs. R3, $p < 0.0001$. DMSO vs. R4, $p < 0.0001$.) $*p < 0.05$, $**p < 0.01$, $***p < 0.001$, n.s. = not significant. **d** Western blot analysis (upper panel) showing the levels of total, chromatin-bound and soluble PCNA in WT and ZFP281 KO ES cells. Histone H3 and MEK2 were used as loading controls. Gray values (lower panel) of PCNA protein bands were determined by Image J and normalized to the loading control. Mean ± SEM from three independent experiments. Two-tailed, unpaired Student's $t$ tests were performed. (Total PCNA, WT vs. ZFP281 KO, $p = 0.0740$; Chromatin-bound PCNA, WT vs. ZFP281 KO, $p < 0.0001$; Soluble PCNA, WT vs. ZFP281 KO, $p < 0.0001$.) $*p < 0.05$, $**p < 0.01$, $***p < 0.001$, n.s. = not significant. **e** Reciprocal immunoprecipitation analyses showing the interaction between ZFP281 and PCNA. The experiments were performed three times with similar results. **f** The representative images of ZFP281/PCNA-PLA foci in DMSO and APH treated ES cells (left panel). PLA using IgG and PCNA antibodies showed no background ($n = 50$). Quantification of the number of PLA puncta per nucleus in DMSO ($n = 222$) and APH ($n = 171$) treated ES cells (right panel). Error bars represent 95% confidence intervals. Two-tailed, unpaired Student's $t$ tests were performed. IgG + PCNA vs. ZFP281 + PCNA, $p < 0.001$; ZFP281 + PCNA (DMSO) vs. ZFP281 + PCNA (APH) $p < 0.0001$.) $*p < 0.05$, $**p < 0.01$, $***p < 0.001$, n.s. = not significant. **g** BrdU re-ChIP assays showing that ZFP281 binds to nascent DNAs at the tested regions. Mean ± SEM from three independent experiments. **h** BrdU IP-qPCR analyses showing BrdU incorporation defects at the *Gata6*, *Pou3f2*, *Kctd1* and *Lhx4* loci in ZFP281 KO cells after release from APH treatment. Mean ± SEM from three independent experiments.

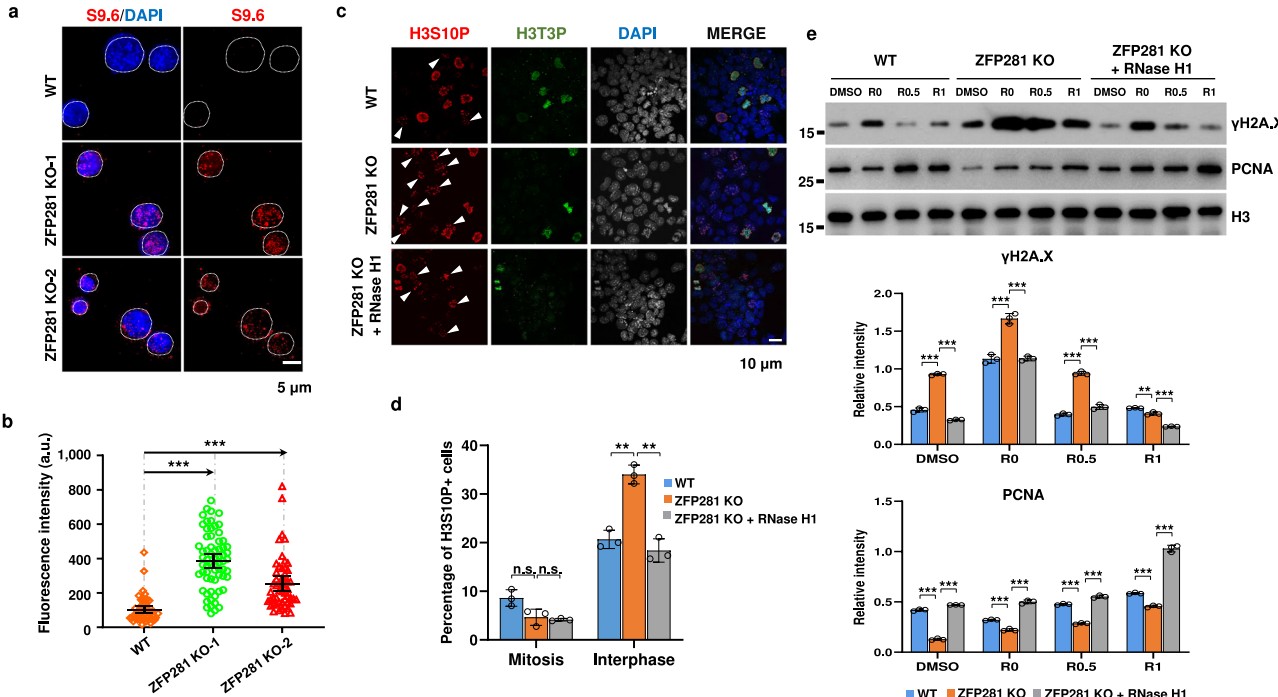

**Fig. 3 ZFP281 depletion leads to aberrant R-loop accumulation. a** Representative immunofluorescence images with the S9.6 antibody showing the levels of R-loop in WT, ZFP281 KO-1, and ZFP281 KO-2 ES cells. DNA was counterstained using DAPI. Three independent experiments show similar results. **b** Fluorescence intensities of R-loop loci in WT ($n = 48$), ZFP281 KO-1 ($n = 66$) and ZFP281 KO-2 ($n = 53$) ES cells. Three independent experiments show similar results. Error bars represent 95% confidence intervals. Two-tailed, unpaired Student's $t$ tests were performed. (WT vs. ZFP281 KO-1, $p < 0.0001$; WT vs. ZFP281 KO-2, $p < 0.0001$.) $*p < 0.05$, $**p < 0.01$, $***p < 0.001$, n.s. = not significant. **c** Immunofluorescence imaging of H3S10P and H3T3P in WT, ZFP281 KO, and ZFP281 KO with RNase H1 overexpression ES cells. White arrows showing the interphase cells positive for H3S10P. **d** Percentage of WT (total $n = 369$), ZFP281 KO (total $n = 380$) and ZFP281 KO with RNase H1 overexpression (total $n = 290$) ES cells in mitosis and interphase. Mean ± SEM from three independent experiments. Two-tailed, unpaired Student's $t$ tests were performed. (Mitosis, WT vs. ZFP281 KO, $p = 0.0574$; ZFP281 KO vs. ZFP281 KO + RNase H1, $p = 0.5526$. Interphase, WT vs. ZFP281 KO, $p = 0.0020$; ZFP281 KO vs. ZFP281 KO + RNase H1, $p = 0.0016$.) $*p < 0.05$, $**p < 0.01$, $***p < 0.001$, n.s. = not significant. **e** Western blot analysis (upper panel) showing the levels of chromatin-bound γH2A.X and PCNA in WT, ZFP281 KO, and ZFP281 KO with RNase H1 overexpression ES cells after release from APH treatment for 0, 0.5 and 1 h respectively. Histone H3 was used as a loading control. Gray values (middle and lower panels) of γH2A.X and PCNA protein bands were determined by Image J and normalized to the loading control. Mean ± SEM from three independent experiments. Multiple $t$ tests were performed. (γH2A.X in WT and ZFP281 KO, DMSO, $p < 0.0001$; R0, $p = 0.0005$; R0.5, $p < 0.0001$; R1, $p = 0.0035$. γH2A.X in ZFP281 KO and ZFP281 KO + RNase H1, DMSO, $p < 0.0001$; R0, $p = 0.0003$; R0.5, $p < 0.0001$; R1, $p = 0.0001$. PCNA in WT and ZFP281 KO, DMSO, $p < 0.0001$; R0, $p = 0.0002$; R0.5, $p < 0.0001$; R1, $p < 0.0001$. PCNA in ZFP281 KO and ZFP281 KO + RNase H1, DMSO, $p < 0.0001$; R0, $p < 0.0001$; R0.5, $p < 0.0001$; R1, $p < 0.0001$.) $*p < 0.05$, $**p < 0.01$, $***p < 0.001$, n.s. = not significant.

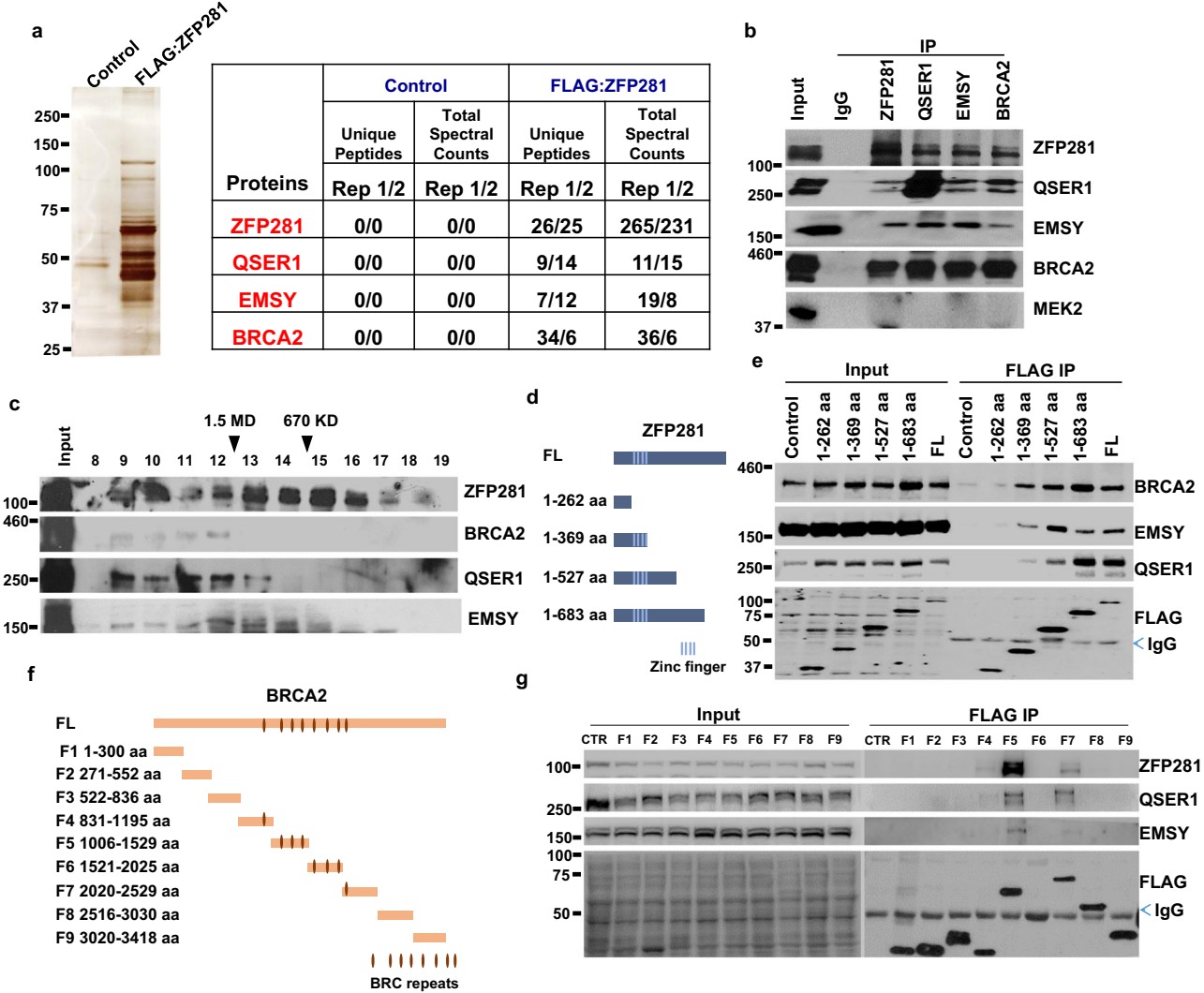

**Fig. 4 ZFP281 interacts with BRCA2. a** Purification and mass spectrometry analyses of the ZFP281-associated proteins. Clonal cell lines expressing FLAG-tagged ZFP281 were generated in 293 Flp-in-TRex cells and the ZFP281 associated proteins were purified using the FLAG-affinity purification method and analyzed by SDS-PAGE, silver staining and mass spectrometry. At least two independent biological repeats with similar results. **b** Confirmation of the interaction of ZFP281 with EMSY, BRCA2, and QSER1 by endogenous immunoprecipitations. **c** Size exclusion chromatography of ES nuclear extracts demonstrated that the ZFP281, BRCA2, EMSY and QSER1 co-elute at ~ 2 MDa (fractions 9–12). **d**, **e** Different ZFP281 truncated proteins were expressed with an N-terminal FLAG tag in 293 T cells, and the interactions of BRCA2, EMSY, QSER1 with these ZFP281 truncated proteins were examined by FLAG immunoprecipitations, followed by western blot analyses. **f**, **g** Different BRCA2 truncated proteins were expressed with an N-terminal FLAG tag in 293 T cells, and the interactions of ZFP281, EMSY, QSER1 with these BRCA2 truncated proteins were examined by FLAG immunoprecipitations, followed by western blot analyses. At least three biological replicates were performed in **b**, **c**, **e** and **g** with similar results.

FLAG-ZFP281 purification products. In addition, we found here that the BRC repeat domain is also essential for the interaction of BRCA2 with ZFP281. Thus, we postulated that ZFP281 and RAD51 might be mutually exclusive in interacting with BRCA2.

**ZFP281 and BRCA2 co-occupy G/C rich promoters in mouse ES cells.** BRCA2 is a key and versatile custodian in maintaining genome integrity through mediating homologous recombination DNA repair, DNA replication, and R-loop resolution[45]. Co-localization of BRCA2 with the DNA polymerase processivity factor PCNA has also been reported previously[46]. Given that ZFP281 loss results in PCNA loading failure and early DNA replication defect through accumulating R-loop, we hypothesized that ZFP281 may function in preventing aberrant R-loop accumulation together with BRCA2. To test our hypothesis, we first carried out BRCA2 ChIP-seq analysis in mouse ES cells, and identified 5,467 significant binding sites, 73.4% of which were

present in transcription start site (TSS) regions (Fig. 5a). A de novo motif search performed under these BRCA2 bound sites demonstrated that the G/C rich sequence was significantly enriched (Fig. 5b), which resembles the previously identified ZFP281 recognition motif[24,25]. Indeed, around 83% (4,550) of BRCA2 sites were also bound by ZFP281 (Supplementary Fig. 7a). Our ChIP-qPCR analyses demonstrated that EMSY and QSER1 also occupied the randomly selected ZFP281 and BRCA2 co-bound regions in mouse ES cells (Supplementary Fig. 7b, c).

Compared to the BRCA2 only regions and the ZFP281 only regions, ZFP281 and BRCA2 co-bound peaks showed a greater enrichment for the TSS signature of H3K4me3 (Fig. 5c and Supplementary Fig. 8). To further characterize the ZFP281 and BRCA2 co-bound peaks, we clustered the 4,550 co-bound peaks based on the co-occurrence of ZFP281 and BRCA2 with the active mark H3K4me3 and the repressive mark H3K27me3. The ZFP281 and BRCA2 co-bound peaks were clustered into two

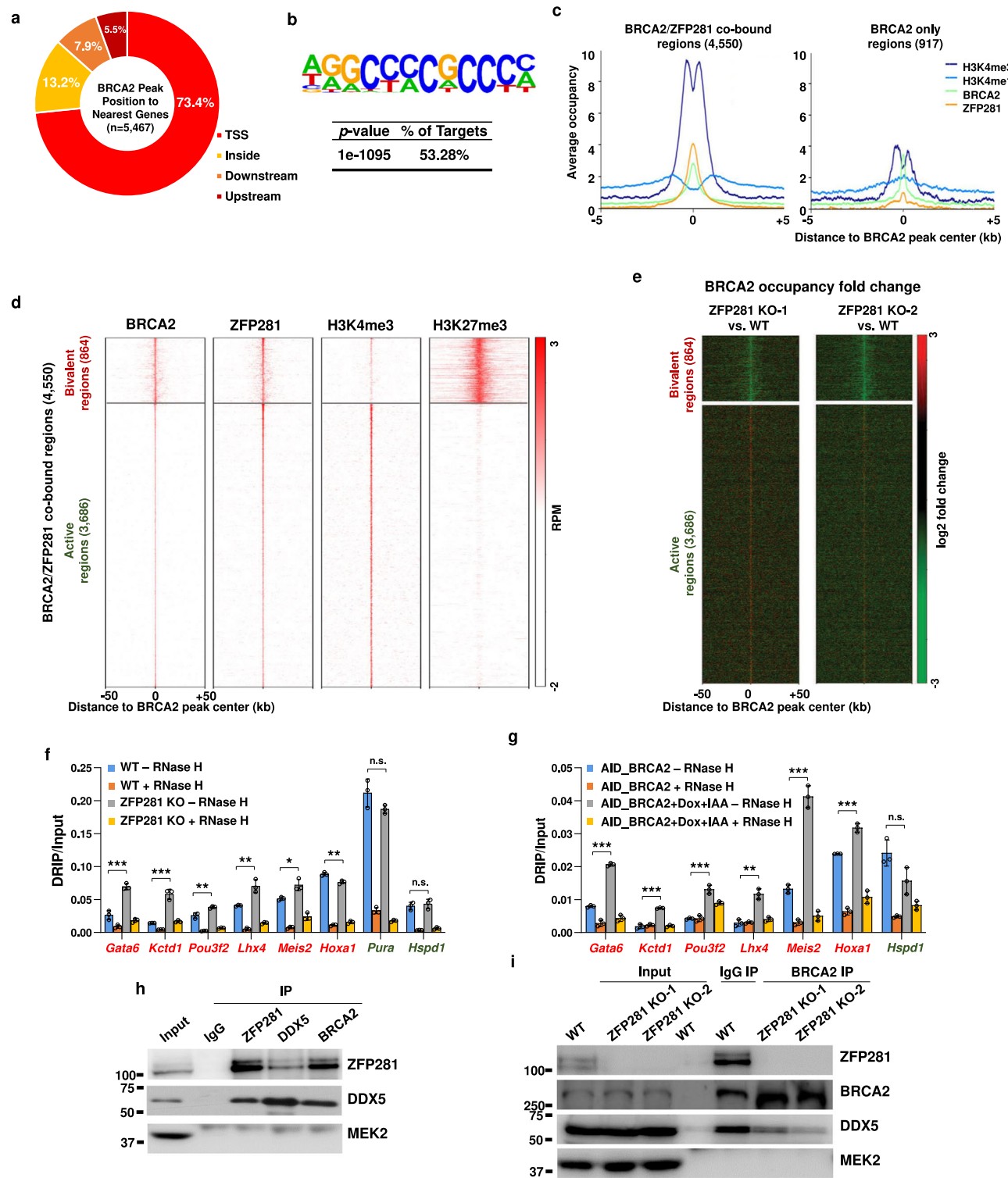

major groups. 3,686 of them were only associated with H3K4me3, and designated as the active group (Fig. 5d). The rest of the peaks had lower H3K4me3 signals, but were highly enriched with H3K27me3, and designated as the bivalent group (Fig. 5d). Functional term analyses indicated that genes involved in metabolic process and stem cell maintenance were enriched in the active group, whereas the bivalent group genes are involved in various developmental processes, such as pattern specification, cell fate commitment and embryonic morphogenesis (Supplementary Fig. 9).

**ZFP281 recruits BRCA2 to the bivalent chromatin in suppressing R-loop accumulation**. It has been shown that both ZFP281 and BRCA2 can directly bind nascent DNA[47,48]. Therefore, we next examined the requirement of ZFP281 and/or BRCA2 for the association of their interacting factors with chromatin. ChIP-seq and -qPCR analyses in control and ZFP281 knockdown cells demonstrated that depletion of ZFP281 specifically reduced the enrichment of BRCA2 to their target sites at the H3K27me3-marked bivalent group regions (Supplementary Fig. 10a, b). We further performed BRCA2 ChIP-

**Fig. 5 ZFP281 is required for the recruitment of BRCA2 to the bivalent regions in mouse ES cells. a** Pie chart showing the percentage of BRCA2 peaks overlapping with a transcription start site (TSS), residing within a gene (inside), or upstream or downstream of the nearest gene. **b** G/C rich sequence is overrepresented in BRCA2 peaks. Statistical significance of the over-representation and the percentage of the G/C rich sequence in BRCA2 peaks are shown. $p$ value from hypergeometric test. **c** Average occupancy plots of ZFP281, BRCA2, H3K4me1 and H3K4me3 at the BRCA2 and ZFP281 co-bound regions (left panel) and the BRCA2 only regions (right panel) in mouse ES cells. Shown are ±5 kb of the center of BRCA2 peaks. **d** Heat maps of the binding profiles in mouse ES cells for BRCA2, ZFP281, H3K4me3, and H3K27me3 are shown at the BRCA2 and ZFP281 co-bound regions, which were partitioned into two groups: the H3K4me3 and H3K27me3 co-bound (bivalent) group, and the H3K4me3 only (active) group. Shown are ±50 kb of the center of BRCA2 peaks. **e** BRCA2 occupancy log2 fold change after ZFP281 KO was measured at the BRCA2 and ZFP281 co-bound regions. Shown are ±50 kb around the center of BRCA2 peaks. **f, g** DRIP-qPCR showing that the levels of R-loop are reduced at the tested bivalent regions (labeled with red color) after ZFP281 knockout (**f**) or AID mediated BRCA2 knockdown (**g**), while remains unchanged at the tested active *Pura* and *Hspd1* loci (labeled with green color). DRIP-qPCR analyses in samples pre-treated with RNase H were used as negative controls. Mean ± SEM from three independent experiments. Multiple $t$ tests were performed. (**f**, *Gata6*, $p = 0.0009$; *Kctd1*, $p = 0.0006$; *Pou3f2*, $p = 0.0092$; *Lhx4*, $p = 0.0069$; *Meis2*, $p = 0.0170$; *Hoxa1*, $p = 0.0051$; *Pura*, $p = 0.1255$; *Hspd1*, $p = 0.5828$. **g** *Gata6*, $p < 0.001$; *Kctd1*, $p = 0.0002$; *Pou3f2*, $p = 0.0003$; *Lhx4*, $p = 0.0012$; *Meis2*, $p = 0.0002$; *Hoxa1*, $p = 0.0005$; *Hspd1*, $p = 0.0611$.) ∗$p < 0.05$, ∗∗$p < 0.01$, ∗∗∗$p < 0.001$, n.s. = not significant. **h** Reciprocal immunoprecipitation analyses showing the interaction between ZFP281 and DDX5. **i** Immunoprecipitation analyses showing that the interaction between BRCA2 and DDX5 was compromised after ZFP281 knockout. The experiments in **h** and **i** were performed three times show similar results.

seq analyses in ZFP281 KO mouse ES cells and confirmed substantial reduction of BRCA2 enrichments over the bivalent group regions, but not the active group regions (Fig. 5e). EMSY and QSER1 also required ZFP281 for their localization at the bivalent group regions (Supplementary Fig. 10c, d). In contrast, BRCA2 was largely dispensable for the occupancy of ZFP281 (Supplementary Fig. 10e). Western blot analyses indicated that knockdown of the individual factor had no major effects on the protein levels of the other factors (Supplementary Fig. 10f), indicating that the reduced chromatin occupancies observed above were not due to a reduction in protein levels.

Since both ZFP281 and BRCA2 are involved in preventing R-loop accumulation, we next analyzed R-loop levels at the ZFP281 and BRCA2 co-bound regions by DNA-RNA immunoprecipitation (DRIP) assay in the cells depleted of ZFP281 or BRCA2. R-loops were highly enriched at the active group regions, but hardly detected at the bivalent group regions (Fig. 5f, g and Supplementary Fig. 11a, b). R-loop levels were not affected after deletion of the subunit of the H3K27me3 depositor PRC2 (Supplementary Fig. 11a)[40,49]. However, DRIP-qPCR analyses revealed that either ZFP281 depletion or degron-mediated BRCA2 degradation led to significant increases in R-loop levels at the bivalent group regions, including *Gata6*, *Kctd1* and *Lhx4*, but had no obvious effect on the R-loop levels at the tested active group regions, such as *Pura* and *Hspd1* (Fig. 5f, g and Supplementary Fig. 11b). In addition, BrdU IP-qPCR analyses revealed that replication defects occurred at the bivalent, but not the active regions, after ZFP281 knockout. RNase H1 overexpression was able to rescue the replication defects caused by ZFP281 knockout at the bivalent regions (Supplementary Fig. 11c).

We also found that R-loop removal by RNase H1 overexpression elevated the occupancies of ZFP281 and BRCA2 at these bivalent regions (Supplementary Fig. 12a, b), accompanied by slight upregulation of the expression levels of these bivalent genes (Supplementary Fig. 12c). In contrast, the bivalent marks remained nearly unchanged after RNase H1 overexpression, which is consistent with the previous study and possibly due to low turnover rates of the epigenetic marks during the short window of the experiments (Supplementary Fig. 12d, e)[49]. Our results suggested that ZFP281 and BRCA2 might both function in preventing R-loop accumulation at the subset of bivalent regions in mouse ES cells, and that reciprocally R-loop could also restrict the binding of ZFP281 and BRCA2 to these bivalent regions.

**ZFP281 is required for the interaction between BRCA2 and DDX5.** BRCA2 interacts with the DEAD-box helicase DDX5 and

stimulates its unwinding activity to resolve R-loop[50]. The interplay between PRC2 and DDX5 has also been observed by different groups[51,52]. Here we found that ZFP281 was also able to interact with DDX5, besides BRCA2, by immunoprecipitation assays (Fig. 5h). In addition, we found that the interaction between BRCA2 and DDX5 was substantially reduced after ZFP281 KO (Fig. 5i). Thus, our results suggest that ZFP281 might be required for the interaction between BRCA2 and DDX5, and thus preventing R-loop accumulation.

**PRC2 is also required for recruiting BRCA2 to the bivalent chromatin.** Given that ZFP281 is essential for the loading of BRCA2 to the bivalent group regions, we then further examined the requirement of H3K27me3 for the genomic occupancies of ZFP281 and BRCA2. Western blot analyses indicated that disruption of the PRC2 subunit EED did not lead to obvious change in the protein levels of ZFP281 and BRCA2 (Supplementary Fig. 13a). However, ChIP-seq and -qPCR analyses revealed that EED loss almost completely abolished the occupancies of BRCA2, but not ZFP281, at the bivalent group regions (Fig. 6a–c). We also observed similar phenomena in mouse ES cells treated with the selective PRC2 inhibitor EI1, which inhibits PRC2 methyltransferase activity and reduces H3K27 methylation without affecting the chromatin binding capability of PRC2 (Supplementary Fig. 13b–d). An interaction between ZFP281 and SUZ12, another core component of PRC2, has previously been reported[53]. Our endogenous immunoprecipitations demonstrated that SUZ12 was also able to interact with BRCA2, in addition to ZFP281 (Fig. 6d). Thus, the recruitment of BRCA2 to the bivalent group regions could also be mediated by the direct interaction between BRCA2 and PRC2.

**ZFP281 and BRCA2 required for the proper activation of PRC2-bound genes during ES cell differentiation.** Given that ZFP281 is essential for the loading of BRCA2 to the bivalent group regions, we also examined whether the expression levels of these ZFP281 and BRCA2 co-bound bivalent genes were affected in the absence of ZFP281. Box plot analyses showed that ZFP281 KO did not seem to have major effects on the expression of this group of genes (Supplementary Fig. 14a). Bivalent genes are silenced in ES cells but poised for activation during stem cell differentiation process. The repressive mark H3K27me3 is dislodged from the bivalent domains during ES cell differentiation[54]. In accordance with expectation, both ZFP281 and BRCA2 were largely disassociated from the bivalent chromatin after neuronal induction of ES cell differentiation by retinoic acid (RA)

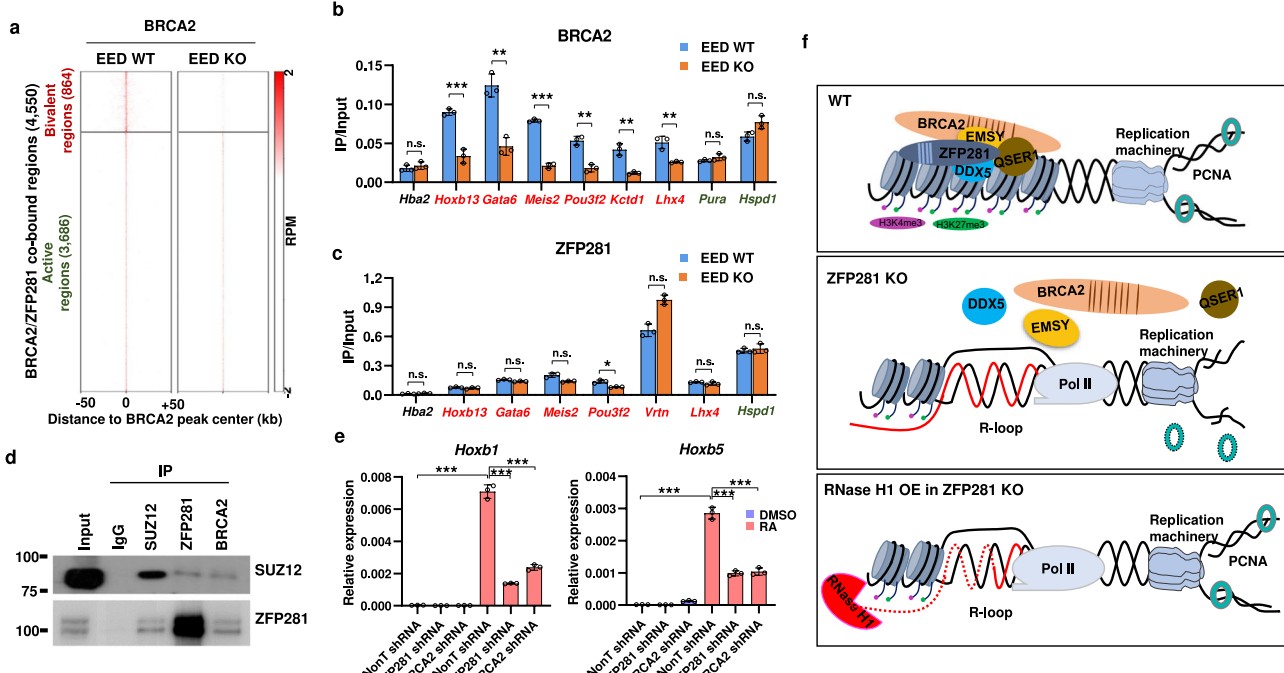

**Fig. 6 BRCA2 also requires PRC2 for its binding to the bivalent chromatin. a** Heat maps of the BRCA2 binding profile in WT and EED KO ES cells. Shown are ±50 kb of the center of BRCA2 peaks. **b** ChIP-qPCR showing that the occupancies of BRCA2 are reduced at the tested bivalent regions (labeled with red color) in EED KO ES cells, while remains unchanged at the tested active *Pura* and *Hspd1* loci (labeled with green color). **c** ChIP-qPCR showing that the occupancies of ZFP281 remains unchanged after EED KO. **b**, **c** The *HEMO* gene (*Hba2*) serves as a negative control for ChIP-qPCR. Mean ± SEM from three independent experiments. Multiple *t* tests were performed. (**b**, *Hba2*, $p = 0.3989$; *Hoxb13*, $p = 0.0006$; *Gata6*, $p = 0.0019$; *Meis2*, $p < 0.0001$; *Pou3f2*, $p = 0.0012$; *Vrtn*, $p = 0.0023$; *Lhx4*, $p = 0.0055$; *Pura*, $p = 0.1818$; *Hspd1*, $p = 0.0564$. **c** *Hba2*, $p = 0.1929$; *Hoxb13*, $p = 0.3279$; *Gata6*, $p = 0.1113$; *Meis2*, $p = 0.1445$; *Pou3f2*, $p = 0.0102$; *Vrtn*, $p = 0.0657$; *Lhx4*, $p = 0.2136$; *Hspd1*, $p = 0.5652$.) ∗$p < 0.05$, ∗∗$p < 0.01$, ∗∗∗$p < 0.001$, n.s. = not significant. **d** Endogenous immunoprecipitation analyses showing the interaction of SUZ12 with ZFP281 and BRCA2. Three independent biological repeats with similar results. **e** RT-qPCR showing that the induction of *Hoxb5* and *Hoxb1* by RA is impaired in ZFP281 KO ES cells. Results shown are technical replicates from representative biological replicates. Mean ± SEM from three independent experiments. Two-tailed, unpaired Student's *t* tests were performed. (*Hoxb1* and *Hoxb5* in NonT shRNA, DMSO vs. RA24, $p < 0.0001$. *Hoxb1* in RA24, NonT shRNA vs. ZFP281 shRNA, $p < 0.0001$; NonT shRNA vs. BRCA2 shRNA, $p < 0.0001$. *Hoxb5* in RA24, NonT shRNA vs. ZFP281 shRNA, $p < 0.0001$; NonT shRNA vs. BRCA2 shRNA, $p = 0.0001$.) ∗$p < 0.05$, ∗∗$p < 0.01$, ∗∗∗$p < 0.001$, n.s. = not significant. **f** Cartoon model showing that ZFP281 functions together with BRCA2 to facilitate DNA replication through preventing R-loop accumulation in the bivalent chromatin (upper panel). ZFP281 depletion impairs the binding of BRCA2 to the bivalent chromatin, leads to aberrant R-loop accumulation and subsequent replication defects (middle panel). RNase H1 overexpression in ZFP281 knockout cells is able to rescue the DNA replication defects caused by ZFP281 knockout (lower panel).

(Supplementary Fig. 14b, c). To further investigate the potential roles of ZFP281 and BRCA2 at the activation of the bivalent genes during differentiation, we depleted ZFP281 and BRCA2 followed by RA treatment. RT-qPCR analyses indicated that the activation of *Hoxb1*, *Hoxb4*, and *Hoxb5* by RA was compromised after ZFP281 or BRCA2 depletion (Fig. 6e and Supplementary Fig. 14d). In summary, our data indicated that ZFP281 and BRCA2 are also required for the full activation of the PRC2-bound genes during differentiation.

## Discussion

R-loops are widely distributed across the genome. Unscheduled accumulation of R-loop poses a major threat to genome stability. However, it is not well understood whether and how prevention of persistent R-loop could occur at specific genomic region to maintain genome stability. In this study, we found that ZFP281 depletion leads to unscheduled accumulation of R-loops over the bivalent regions, affecting chromatin-bound PCNA levels and DNA replication at early S phase. Removal of R-loops by RNase H1 overexpression is able to rescue the DNA replication defects caused by ZFP281 depletion. ZFP281 interacts with BRCA2, and

is required for the recruitment of BRCA2 to the bivalent chromatin regions to prevent R-loop accumulation (Fig. 6f). In summary, our study suggested a mechanism by which BRCA2 is recruited to the bivalent chromatin regions to prevent persistent R-loops.

We have previously demonstrated that ZFP281 occupies and represses retrotransposons, including long interspersed element-1 (LINE-1)[27]. Retrotransposons, which make up almost half of the mammalian genome, have been reported to be R-loop hot spots[6]. Thus, in addition to its regulation to the R-loop at the subset of bivalent regions, ZFP281 might also be involved in preventing excessive R-loop at the repetitive regions. Indeed, S9.6 antibody staining in WT control and ZFP281 KO mouse ES cells indicated that R-loop levels were substantially increased after ZFP281 loss.

BRCA2 has been previously demonstrated to function in suppressing R-loop accumulation[14,55]. According to our ChIP-seq analyses, despite that BRCA2 and its cofactor ZFP281 are enriched at both the active and the PRC2-bound gene promoters, ZFP281 is only specifically required for the recruitment of BRCA2 to the PRC2-bound chromatin. It has been previously established that BRCA2 selectively binds to single stranded DNA (ssDNA)[48]. Thus, one possible explanation is that highly dynamic

transcription process at the active regions generates ssDNA structures that favor BRCA2 binding; while at the repressive PRC2 bound regions BRCA2 requires ZFP281 and PRC2 to facilitate its binding to DNA. In addition, whether other members in the Fanconi Anemia pathway, besides BRCA2, are involved in restricting R-loop levels at the bivalent regions is worth further exploration[15,16].

BRCA2 plays a critical role in DNA replication fork protection[45]. PRC2-bound regions tend to be early DNA replication sites in multiple cell types, including mouse ES cells, and the H3K27me3 is rapidly restored at downstream of DNA replication fork[56]. In the present study, we have demonstrated that both ZFP281 and PRC2 are required for the recruitment of BRCA2 at the bivalent chromatin domains to prevent R-loop accumulation, providing a detailed mechanism for maintaining genome stability by BRCA2 through R-loop suppression.

BRCA2 homozygous mutant mice show severe developmental defect or even embryonic lethality[57]. Accumulation of R-loops could be a source of DNA damage, eventually leading to premature differentiation, although ES cells make more robust response to DNA damage to ensure genome integrity[56,58]. Our study links ZFP281-BRCA2's R-loop suppression function with PRC2 at replicating bivalent chromatin. Disruption of ZFP281 or BRCA2 is also detrimental for the further activation of the bivalent genes during stem cell differentiation. Therefore, it is possible that the cross talk between ZFP281-BRCA2 and PRC2 is essential in maintaining proper chromatin structures, and the disruption of their interplay could contribute to early developmental defects and cancer pathogenesis.

## Methods

**Mouse ES cell culture**. Mouse ES cell lines V6.5, EED WT, EED KO and RNase H1 inducible KH2 were cultured in N2B27 medium supplemented with 2i/LIF in 0.1% gelatin-coated tissue culture flask[59]. For EI1 treatment, EI1 powder (Cayman Chemical) was dissolved in DMSO as 10 mM stock solution. The cells were treated with 10 μM of EI1 or DMSO as control and incubated for 48 hrs. For differentiation, mouse ES cells were treated with RA for 24 hrs and harvested for RT-qPCR analysis[60]. All cells were maintained at 37 °C under 5% CO2.

**CRISPR-Cas9 guided knockout**. ZFP281 NT and CT sgRNA oligos were cloned into lentiCRISPR v2. Lentiviral particle preparation and infection were performed as below described[61]. Briefly, around 70% confluent 293 T cells in 150 mm tissue culture plate were co-transfected with 8 μg of the lentiCRISPR v2 sgRNA constructs, 6 μg of psPAX2 packaging plasmids and 2 μg of pMD2.G envelope plasmids using X-tremeGENE HP (Roche). The media were replaced with fresh DMEM supplemented with 10% fetal bovine serum after 16 hrs of transfection. The lentiviral supernatants were collected 48 and 72 hrs after the transfection, filtered through 0.45 μm filters and concentrated at 50,000 × g for 2 hrs. ZFP281 NT sgRNA lentiviral particles were mixed with ZFP281 CT sgRNA-1 and sgRNA-2 lentiviral particles, respectively. Mouse V6.5 ES cells were infected with concentrated lentiviral particles with polybrene (Sigma) at the concentration of 8 μg/ml. V6.5 cells were selected with 2 μg/mL puromycin for 48 hrs in 2i/LIF medium. The selected cells were maintained in 2i/LIF medium without puromycin until cell clones were ready to be picked. The clones were screened with PCR, and confirmed by TA cloning and sequencing.

**Lentivirus mediated RNAi**. Mouse ZFP281 shRNA construct was described previously[26]. BRCA2 and QSER1 shRNA constructs were cloned into the pLKO.1 vector (Addgene). The Non-targeting shRNA construct (SHC002) was purchased from Sigma. Lentiviral particle preparation and infection were performed as described above. 24 hrs after infection, the ES cells were subjected to selection with 2 ug/mL of puromycin for an additional 48 hrs.

**Auxin-inducible degron (AID) system**. BRCA2 CT sgRNA oligos were cloned into pX330. Homology arms flanking the target site in the last exon of *Brca2* were amplified by PCR from V6.5 genomic DNA. To fuse the AID cassette to the C-terminus of BRCA2, V6.5 mouse ES cells with TIR1-expressing cassette were co-transfected with the pX330 BRCA2 CT sgRNA construct and the homology arms, followed by selection with hygromycin. The transfected cells were maintained in 2i/LIF medium with hygromycin until cell clones were ready to be picked. The clones were screened with PCR, and confirmed by TA cloning and sequencing.

**Antibodies**. Antibody against ZFP281 was described previously (WB, 1:6000)[26]. Antibodies against BRCA2 (WB, 1:2000; IF, 1:800) and QSER1 (WB, 1:5000; IF, 1:800) were generated in-house, Antibodies against ZFP281 (Abcam, ab112047, PLA, 1:200), H3K27me3(Sigma, 07–449, WB:1:2000), EMSY (Abcam, ab123, WB, 1:5000), H3 (Abcam, ab1791, WB, 1:10000), V5 (Abcam, ab9116, WB, 1:5000), SUZ12 (Abcam, ab12073, WB, 1:3000) and H3S10P (Abcam, ab14955, IF, 1:1000) were purchased from Abcam. Antibodies against α-TUBULIN (Sigma, T9026, WB, 1:10000), BrdU (Sigma, RPN202), H3T3P (Sigma, 05–746 R, IF, 1:100) and FLAG (Sigma, F3165, WB, 1:5000) were purchased from Sigma. Antibodies against PCNA (Santa Cruz, sc-25280, WB, 1:1000; PLA, 1:100; IF, 1:200) and γH2A.X (Santa Cruz, sc-517348, WB, 1:1000) were purchased from Santa Cruz. Antibodies against Cyclin A2 (ABclonal, A7632, WB, 1:1000), Cyclin E1 (ABclonal, A14225, WB, 1:800), Cyclin C (ABclonal, A13610, WB, 1:1000) and DDX5 (ABclonal, A11339, WB, 1:1000) were purchased from ABclonal. The S9.6 antibody (ENH001, IF, 1:100) against R-loop was purchased from Kerafast.

**Cell cycle analysis**. Cell cycle analysis were performed using the BeyoClick™ EdU Cell Proliferation Kit with Alexa Fluor 488 (Beyotime, C0071S) according to the manufacture's instruction. Briefly, cells were pulsed with 10 μM EdU for 30 min. Then cells were harvested, washed with PBST (0.5% Tween-20), incubated with RNase A for 30 min at 37 °C, and stained with 50 μg/mL propidium iodide (PI) for 30 min, followed by flow cytometry analysis. CellQuest Pro (BD FACS caliber) software was used for data collection, and FlowJo 7.6.1 was used for data analysis.

**EdU labeling assay**. Click-iT™ EdU Cell Proliferation Kit for Imaging, Alexa Fluor™ 555 dye (C10338, Invitrogen) was used for detecting cells at different stage of S phase after ZFP281 KO, according to the manufacture's protocol.

**Early S phase synchronization and release**. Mouse ES cells were treated with the DNA polymerase inhibitor aphidicolin (APH) to inhibit DNA replication, and grown for 16 hrs in 0.3 μM Aphidicolin ES media, then released from the replication block by culturing in APH-free media for 0, 1, 2, 3, 4 hrs respectively.

**Proximity ligation assay**. Proximity ligation assay was performed on mouse ESCs using the Sigma-Aldrich Duolink In Situ PLA manufacturer's protocol (Sigma, DUO92101). In brief, cells were seeded on coverslips and treated with DMSO or APH (0.3 μM, 16 h). Fixed cells were permeabilized and blocked. The primary antibodies against ZFP281 (ab112047, 1:200) and PCNA (SC-25280, 1:100) were applied, and the cells were incubated with two PLA probes, anti-Rabbit PLUS (target anti-ZFP281 or IgG) and anti-mouse MINUS (target anti-PCNA). Hybridization, ligation and amplification were carried out by the manufacturer's instructions. Coverslips were mounted with mounting media containing DAPI. For the fluorescent images, Zeiss LSM 800 with 63× oil immersion objectives were used and analyzed using ImageJ software. Quantification of PLA signal was performed on z-stack images.

**FLAG immunoprecipitation, purification and mass spectrometry analysis**. FLAG-tagged full length and truncated ZFP281 were cloned into pcDNA5/FRT-TO vector (Invitrogen) with N terminal FLAG tag. The expression plasmids were transfected into HEK293T cells for 48 hrs followed by FLAG immunoprecipitations using ANTI-FLAG M2 affinity gel (Sigma).

For FLAG purification, Flp-In-TRex system was used to generate the FLAG-ZFP281 expressing cell line[41]. The pcDNA5/FRT-ZFP281 expression plasmids were transfected into 293 Flp-in-TRex cells and selected for with 100 ug/mL hygromycin. The expression of FLAG-ZFP281 proteins was induced with 2 μg/mL doxycycline for 48 hrs. Nuclear extracts were prepared using the high salt buffer (20 mM HEPES [pH 7.4], 420 mM NaCl, 1.5 mM MgCl₂, 0.2 mM EDTA, 10% glycerol, 0.5 mM DTT, protease inhibitor cocktail). After centrifugation, the balance buffer (20 mM HEPES [pH 7.4], 1 mM MgCl2, 10 mM KCl) was added to the supernatant to make the final NaCl concentration 300 mM. Nuclear extracts were subjected to FLAG-affinity purification in the presence of benzonase (Sigma). The purification products were analyzed by SDS-PAGE and silver staining. Trichloroacetic acid-precipitated protein mixtures from purifications were digested with endoproteinase Lys-C and trypsin (Roche) and subjected to Mass Spectrometry analysis.

**Immunoprecipitations and Western blotting**. Cells were lysed in high salt lysis buffer containing 420 mM NaCl in the presence of protease inhibitor cocktail (Sigma) for 30 min at 4 °C with gentle rotation. After centrifugation, the balance buffer (20 mM HEPES [pH 7.4], 1 mM MgCl2, 10 mM KCl) was added to the supernatant to make the final NaCl concentration 300 mM. The lysate was then incubated with antibodies and protein A beads overnight at 4 °C. The beads were spun down and washed three times with wash buffer before boiling in SDS loading buffer. Proteins were resolved in SDS-PAGE gels and transferred to polyvinylidene fluoride (PVDF) membrane. Primary antibodies used were incubated overnight at 4 °C. HRP-conjugated secondary antibodies (Invitrogen, 101023) were used at a dilution of 1:5000. ECL substrate (Millipore) was applied to the membrane for imaging by autoradiography. Images were collected by BioRad Imager. At least

three biological replicates were performed for each experiment. Shown are the representative results from the independent biological replicates.

**Size exclusion chromatography.** Nuclear extract was subjected to high resolution Superose 6 increase 3.2/300 column (GE Healthcare) with size exclusion buffer (40 mM HEPES [PH 7.5], 350 mM NaCl, 10% glycerol and 0.1% Tween-20). Fractions were resolved in SDS-PAGE gels, followed by Western blotting.

**Quantitative RT-PCR.** Total RNA was isolated with RNeasy kit (Qiagen), treated with RNase free DNase I (New England Biolabs), and repurified with RNeasy column. cDNAs were synthesized with the PrimeScript™ RT Master Mix (TaKaRa). The expression levels were measured with iTaq™ Universal SYBR® Green Supermix (Bio-Rad) on CFX96 (Bio-Rad). The relative expression levels of genes of interest were normalized to the expression of the housekeeping gene *Actin*. At least three biological replicates were performed for each experiment. Shown are the representative results from the independent biological replicates. Primer sequences are available in Supplementary Table.

**Immunostaining.** Cells were grown on 1% gelatin-coated coverslips, rinsed with PBS, fixed in cold methanol for 10 min at −20 °C, then incubated in cold acetone for 1 min. The coverslips were then quickly washed with 4 × saline sodium citrate (SSC) buffer three times and incubated for 30 min in 4 × SSC buffer. For S9.6 staining, mouse ES cells were trypsinized into single cells, treated with 75 mM KCl hypotonic solution, and then fixed in fixative (methanol/acetic acid glacial 3:1) for 30 min at 4 °C[62]. Spreading was performed on a coverslip to improve the detection of R-loop.

3% BSA buffer was used to block nonspecific interaction before incubating with primary antibodies. Primary antibodies were incubated overnight at 4 °C. After three times of washes with PBST, coverslips were incubated at room temperature with secondary antibodies (anti- mouse IgG Alexa fluor 488, Invitrogen, A-11001, 1:1000; anti- rabbit IgG Alexa fluor 546, Invitrogen, A-11035, 1:1000) for 1 hr, followed by three times of washes with PBST. Cover glasses were mounted on slides with Vectashield mounting media containing DAPI. For the fluorescent images in Fig. 2a and b, Zeiss LSM 880 with Airyscan and GaAsp detectors was used for both conventional confocal and Airyscan superresolution mode with a 63×, Plan Apochromat (1.4 NA) oil objective. For the rest of the fluorescent images, Zeiss LSM 700 was used. At least three biological replicates were performed for each experiment. Shown are the representative results from the independent biological replicates.

**DNA:RNA immunoprecipitation (DRIP).** Cells were digested with proteinase K digestion buffer (50 mM Tris-HCl [pH 8.0], 10 mM EDTA, 0.5% SDS) containing proteinase K at 55 °C overnight, followed by phenol/chloroform extraction and ethanol precipitation. DNA was fragmented using a combination of restriction enzymes (HindIII, BsrGI, XbaI, EcoRI and SspI) in buffer 2.1 overnight at 37 °C. For RNase H-treated control samples, DNA was incubated with RNase H (NEB) overnight at 37 °C and purified as described above. For DRIP, fragmented DNA was incubated with the S9.6 antibody and protein A agarose beads in binding buffer (10 mM NaPO4 [pH 7.0], 140 mM NaCl, 0.05% Triton X-100) overnight at 4 °C. Beads were washed twice with binding buffer and binding buffer containing 330 mM NaCl, then bound DNA was eluted with elution buffer (50 mM Tris-HCl [pH 8.0], 10 mM EDTA, 0.5% SDS) containing Proteinase K for 45 min at 55 °C. The immunoprecipitated and input DNAs were purified and used as templates for qPCR. At least three biological replicates were performed for each experiment. Shown are the representative results from the independent biological replicates. Primer sequences are available in Supplementary Table.

**Chromatin-bound proteins isolation.** Cells were harvested and washed by cold PBS. Cells were resuspended in buffer A (10 mM HEPES [pH 7.9], 10 mM KCl, 1.5 mM MgCl2, 0.34 M sucrose, 10% glycerol, 1 mM DTT) with fresh added proteinase inhibitors. Add Triton X-100 to final concentration to 0.1% in cell suspension, then incubated on ice for 5 min. Centrifuge 1,300 x g for 5 min at 4 °C to collect nuclei pellet. Nuclei were washed once in buffer A, and then lysed in no-salt buffer B (0.2 mM EGTA, 3 mM EDTA and 1 mM DTT, protease inhibitors as described above). Following incubated on ice for 30 min, the insoluble chromatin was collected by centrifugation. The final chromatin pellet wash by buffer B before western blot.

**ChIP, ChIP-seq library preparation.** ChIP assays were performed as described[60]. Briefly, a total of $5 \times 10^7$ cells were cross-linked in PBS containing 1% paraformaldehyde for 10 min at room temperature, cross-linking was quenched by glycine. Fixed chromatin was sonicated into 200–800 bp fragments (Bioruptor, Diagenode) in ChIP lysis buffer (10 mM Tris-HCl [pH 8.0], 100 mM NaCl, 1 mM EDTA, 0.5 mM EGTA, 0.1% Na-Deoxycholate, 0.5% N-lauroylsarcosine) supplemented with protease inhibitor cocktail (Sigma). Chromatin extracts were incubated with specific antibody and protein A agarose beads at 4 °C overnight. Immunoprecipitates were washed with RIPA buffer (50 mM HEPES-KOH [pKa 7.55], 500 mM LiCl, 1 mM EDTA, 1.0% NP-40, 0.7% Na-Deoxycholate) for

five times and TE once. After the final wash, DNA was eluted and reverse-crosslinked at 65 °C. DNA was then purified and used as a template for qPCR or for ChIP-seq library preparation. At least three biological replicates were performed for each experiment. Shown are the representative results from the independent biological replicates. ChIP-seq libraries were prepared with NEB Next sample prep kit for the further next generation sequencing. Primer sequences are available in Supplementary Table.

**Next generation sequencing analysis.** BRCA2 ChIP-seq in V6.5, EED WT, EED KO, V6.5 (NonT shRNA), V6.5 (ZFP281 shRNA), ZFP281 WT and ZFP281 KO data were generated in this study and have been deposited in the GEO database [https://www.ncbi.nlm.nih.gov/geo/], with accession number GSE127262. Other data sets come from previously published studies. H3K4me1 ChIP-seq data from GEO accession number GSE24164[63]; H3K4me3 and H3K27me3 ChIP-seq data from GEO accession number GSE12241[64]; ZFP281 ChIP-seq data from GEO accession number GSE77115 [26]. Clean reads were aligned to the Mus_musculus genome (UCSC genome, mm9) using Bowtie2 (v2.2.5) allowing uniquely mapping reads only[65].

Peak calling was performed with MACS2 (v2.1.1)[66]. For BRCA2 peaks, associated control samples were used to determine statistical enrichment at $p < 1e-5$ and FDR < 0.05. For ZFP281 peaks, associated control samples were used to determine statistical enrichment at $p < 1e-8$ and FDR < 0.05. Co-bound peaks between BRCA2 and ZFP281 peaks were determined if the peak regions of two peaks are overlapping each other. Peak annotating nearest genes and binding to TSS regions were performed with ChIPseeker[67]. The motif analysis was performed by Homer (v4.9) with program 'findMotifsGenome.pl'.

For the heat maps of the ChIP-seq enrichment profiles, the BRCA2 and ZFP281 co-bound regions were divided into two groups, which were the co-bound regions with H3K4me3 only and the co-bound regions with both H3K4me3 and H3K27me3. Each row showed one peak and regions were centered on the BRCA2 peak center. Regions were shown oriented from 5' to 3'corresponding to the orientation of the nearest annotated gene. Regions were spanned 50 kb on both sides. For the Meta plots were produced by deepTools 2.0[68] and regions were spanned 5 kb on both sides. Functional annotation analysis as reported by GREAT[69].

**Reporting summary.** Further information on research design is available in the Nature Research Reporting Summary linked to this article.

## Data availability

ChIP-seq data in this study have been deposited in the in NCBI's Gene Expression Omnibus and are accessible through accession number GSE127262 and GSE77115. Public ChIP-seq accessible in NCBI's Gene Expression Omnibus through GSE24164, GSE12241 were used. Source data are provided with this paper.

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

## Acknowledgements

The authors are grateful to the Lin & Luo lab members for helpful discussion of this study. We thank Drs. Qin Li and Qianwen Sun from Tsinghua University for providing the S9.6 antibody. WT and EED KO ES cell lines were gifts from Drs. Terry Magnuson and Gang Li. Studies in this manuscript were supported by funds provided by National Key R&D Program of China (2018YFA0800100 to C.L.), National Natural Science Foundation of China (32030017, 31970617 to C.L.; 31970626 to Z.L.), Shenzhen Science

and Technology Program (JCYJ20210324133602008 to C.L.; JCYJ20210324133601005 to Z.L.), China Postdoctoral Science Foundation (2021M690616 to Y.W.), and the Fundamental Research Funds for the Central Universities (2242021R20049 to Y.W.).

## Author contributions

Y.W. generated the ZFP281 KO and the AID_BRCA2 ES cell lines, and performed DRIP, ChIP-seq, ChIP-qPCR, RT-qPCR analyses; B.M. and Y.W. performed immunostaining assays, confocal imaging and analyses; G.G. performed PLA assays; Z.L., X.L., and C.L. analyzed the genome-wide sequencing data; Y.W. and Z.C. performed immunoprecipitations; M.F., S.M., and X.Z. performed western blot and RT-qPCR analyses. Z.Z., J.X., S.R., and H.C. provided technical assistance; Z.L. and C.L. designed the research, identified the ZFP281 containing complex, and wrote the manuscript. All authors discussed the results and commented on the manuscript.

## Competing interests

The authors declare no competing interests.
