## [Peer Review File · Nature Communications]

ZFP281-BRCA2 prevents R-loop accumulation during DNA replicationREVIEWER COMMENTS

Reviewer #1 (Remarks to the Author):

This is a priori a nice piece of work in which authors show a new interaction between ZFP281 and BRCA2 with a specific role in the DNA damage response in particular at bivalent chromatin. Authors show a great amount of data to show that a replication defect in cells depleted of ZFP281 pointing out to this defect being mediated by DNA-RNA hybrids, as previously shown for BRCA2-deficient cells. Both proteins would work together to solve the aberrant structure. The study is of interest but it has a number of overinterpretations and results that need to be better supported to claim part of the conclusions.

1) The claim of a ZEBRA complex is not supported with the results provided. Authors should either remove the reference to this complex or show its existence using the appropriate biochemistry tools. Authors just make a pull down overexpressing ZFP281 from a plasmid and further mass-spectrometry analysis. However, we need information how resistant to salt concentrations the putative complex is, whether the interaction is observed in samples in which DNA and RNA are removed (there is no indication about this in M&M, and since they conclude that it is formed at bivalent chromatin, it means that they assume that there is DNA in their preps) and a clean gel where we see the purified components with a minimal idea of stoichiometry between the different members. The conclusion that ZFP mediates the assembly of the complex on the bivalent chromatin, is an overinterpretation, because authors show no experiment addressing this question and no conclusion can be made from ChIP studies on the existence of complexes but only on the accumulation of the factors at the same chromatin region, not necessarily as part of a stable complex or even at the same time during the cell cycle.

2) The data in Figure 1d is not very convincing as cyclin c levels seem to follow the same pattern of cyclin A2 and E in the gel. The quantification does not seem to fully match the gel. The experiment should be repeated and quantifications should be shown as average signal with error bars. Furthermore, quantitative western blots by fluorescent signals rather than ECL would ensure that the loaded proteins are within the linear range of detection and that the signal is not affected by the different amounts of proteins or the kinetics of the ECL reaction. However, to more convincingly claim a cell cycle progression defect, the authors should ideally measure BrdU incorporation profiles by flow cytometry in addition to the cellular quantification of the different S phase patterns in 1e.

3) The data in other western-blot (figure 2d and 3e) should also be properly quantified and reproducible to claim that PCNA levels are reduced upon ZFP281 loss, leading to increased levels of H2AX phosphorylation and to show a slow recovery after APH release and that all these phenotypes are rescued by RNase H overexpression. Additionally, if replication is perturbed in ZFP281 cells, increased fork asymmetry would be expected in DNA combing experiments, as it has been shown in other R-loop accumulating mutants rather than only a slow S phase. This phenotype should be also suppressed by RNase H.

4) Figure 2a shows a single line in a single cell. Given the abundant signal detected in both PCNA and ZFP281 immunofluorescences, it is not surprisingly that they overlap to a certain degree. A more robust data should be presented to claim that these two factors co-localize in the cell, such as a PLA analysis with both antibodies.

5) From Fig. 2d in page it is not possible to conclude that ZFP281 is required for PCNA loading or stabilization on chromatin during DNA replication, since it may well be that an earlier replication event is delayed so that PCNA loading is affected as a consequence but not because ZFP helps PCNA to be loaded. This needs to be better reasoned and clarified.

6) From Figure 2g, the authors state that the BrdU recovery is delayed in ZFP281 KO cells. In contrast, it seems that the peak of maximum BrdU incorporation is at the same time or even earlier in ZFP281 KO cells (green line).

7) Figure 3a should be more than a single cell. Indeed, the second cell partially shown in the WT shows much more S9.6 signal arguing that the chosen cell is not a representative image.

8) The authors conclude that ZFP281 recruits BRCA2 and other proteins to form the ZEBRA complex that is important to prevent R-loops at bivalent promoters based on CHIPseq data. However,

-Figure 6f is not cited in the text. Moreover, as a general model, it would be more illustrative to depict the bivalent regions where this phenomenon occurs. Also, drawing certain replisome components and PCNA unloading seems to indicate that the replisome is intact except for PCNA but this is not tested in the paper at all.

-It is unclear the reasoning to validate the ChIPseq data in ZFP281 KO cells (Figure 5f) performing ChIP-qPCR in shZFP281-depleted cells (Supplementary Figure 7) rather than ZFP281 KO cells or for the DRIP in Figure 5g,h.

-It would be expected that EED KO cells should decrease the signal in DRIP experiments in these bivalent promoters in support of this conclusion.

9) In general, the quality of the graphical representation is poor. The error bars are barely distinguishable in figure such as 2e,f,g (also 5g,h). Also, the statistics is missing in all these graphs whereas the authors state significant variations throughout the paper. Do the data correspond to the average between independent experiments? How many? This is not indicated at all in the legends or the methods.

10) BRCA2 is a member of the Fanconi Anemia pathway, which works at damage blocking replication. Since this pathway has been shown to resolve also r loops (Schwab et al, Mol Cell 2015; García-Rubio et al. PloS Genet 2015) it would be interesting to discuss the results obtained in this context.

Minor points

-page 12, line 268 should say Figure 5g,h instead of 6g,h

-Figure S3E should accompany the main Figure 4f to be able to understand it.

-The Methods section should state where confocal microscopy was used, such as in Figure 2A, where Airyscan imaging is said to have been used in the Results section.

-line 227, page 10. A reference should be cited to indicate the previously identified ZFP281 recognition motif.

-The English language should be revised. Some examples of incorrect spelling/grammar include:

-Chromatin is a not countable noun. Thus, 'chromatins' is not appropriate.

-line 81: 'a' is missing before 'specific regulatory mechanism'

-line 90 page 4, 'previously studies' should be 'previous studies'

Reviewer #2 (Remarks to the Author):

R-loops are unusual but important nucleic acid structures that regulate gene expression and if not tightly regulated can pose serious threats to genome integrity. The authors focused on embryonic

stem cells and used cellular and genomic approaches to show that ZFP281 cooperates with BRCA2 in preventing R-loop accumulation to promote DNA replication. They used biochemical purifications that led to the identification of ZFP281 and BRCA2 in a ZEBRA complex. They further show that the enrichment of BRCA2 at G/C-rich promoters requires both ZFP281 and PRC2, which if disrupted leads to perturbations in PCNA loading and replication defects. This is an interesting story and highly relevant to both the DNA repair and R-loop fields, but the authors need to address the following points:

1. The authors suggest that ZFP281 and BRCA2 might both function in preventing R-loop accumulation at the subset of bivalent regions in mouse ES cells but fail to explain why a global increase is there if the function is only at specific genomic loci.
2. Can the authors tie the R-loop data to bivalent chromatin? E.g. would RNase H1 expression reduce binding of BRCA2 and other ZEBRA components to bivalent domains? Would RNase H1 overexpression change the expression of PRC-2 bound genes? How does ZEBRA resolve R-loops? How does the R-loop resolution lead to epigenetic changes at these specific loci?
3. It is often not clear if the authors performed biological independent replicates or technical repeats, e.g. no error bars in Fig1b.
4. I am not convinced that PCNA in chromatin was significantly reduced in the absence of ZFP281 in Fig.2d. This is complicated by variations in loading and the authors need to provide quantification after normalisation and also show PCNA protein levels in the soluble nucleoplasmic fraction which should show a corresponding increase in PCNA.
5. It is not clear in Fig1d that ZFP281 KO exhibits an effect on cyclin E levels. Can the authors provide quantification of three biological repeats and stats?
6. The previous comment applies to multiple figures in the manuscript such as Fig2c and Fig2d. Also, I am not sure if the current data supports the conclusion that RNase H1 overexpression could partially rescue the delay of PCNA re-loading to chromatin and abrogate the aberrant increase of γ H2A.X after APH release in ZFP281 KO cells (Fig. 3e). There is no quantification nor statistical analysis.
7. The interaction between ZFP281 and PCNA in Fig. 2a requires validation by co-IP or proximity ligation assays.
8. Fig2g - I can't see a delay in recovery. I can see a reduction. Could you comment further on this?
9. The R-loop data in Fig3a need confirmation by an orthogonal method such as DRIP-qPCR
10. Fig3d - Stats and error bars are missing. Should RNase H1 over-expression increase the number of mitotic cells, as the knockout cells are not going through mitosis due to aberrant accumulation of R-loops? Can the authors comment on this discrepancy?
11. Fig3e - the images are not convincing. The authors need to provide an RNase H blot and quantification of PCNA and γ H2A.X signal intensity with stats.
12. Fig4b - the authors need a negative control: i.e. a blot of a protein that ZEBRA components do not bind to.
13. Fig5f - Can the authors explain what KO-1/WT and KO-2/WT mean? The reduction is also not obvious. A better-quality image and a quantification are needed in this figure.
14. Fig5gh - Stats are again missing. Could you also highlight in the figure which primers are 'active' and which ones are 'bivalent'?
15. There are multiple typos, and the manuscript will benefit from a careful proofreading, e.g:
 - a. Line 90: previously studies
 - b. Line 144: for different time

Reviewer #3 (Remarks to the Author):

The manuscript by Wang et al described a ZFP281 and BRCA2-containing protein complex ZEBRA, providing a potential explanation for the replication defects and increased R-loop formation in ZFP281 KO cells. ZFP281 depletion leads to R-loops accumulation, results in delayed PCNA loading and DNA replication defects, which are also observed in BRCA2 KO cells. Genome-wide occupancy analyses

revealed that BRCA2 is highly enriched at G/C-rich promoters and that ZFP281 as well as PRC2 are required for BRCA2 localization at bivalent chromatin. Thus, the ZEBRA complex appears to be required for preventing R-loops and ensures proper progression of DNA replication. The data are of high quality and the conclusion links the replication defects of ZFP281 KO cells to the function of BRCA2.

Major concerns:

1. What are R-loops distributions in ZFP281 and BRCA2 KO cells? What is the percentage of overlapping? Is R-loops only limited to bivalent chromatin? In this case, PRC2 KO cells should also have increased accumulation of R-loops.
2. What is the relationship of replication defects in ZFP281 KO cells and the failure of activating of bivalent genes in ZFP281 KO cells upon differentiation?

Specific comments:

1. Figure 1a, which gRNAs are in ZFP281 KO1 and KO2 cells?
2. Figure 1d, are changes in these cyclins in protein levels or RNA levels? Please provide a quantitative measurement.
3. Figure 1e, please explain: S phase cell is reduced but early S phase cell is increased?
4. Lane 131, please explain: however, the EdU positive ratios of entire S phase were comparable between WT and ZFP281 KO cells. This seems contradicted with Figure 1e.
5. How are the R0-R4 related to cell cycle progression?
6. Supplementary Fig. 2 is not appropriately cited (page 8).
7. Figure 4c, label the molecular size of gel filtration column.
8. Figure 4, draw a diagram to indicate the interactions between different proteins (domains).
9. Citation for "co-localization of BRCA2 with the DNA polymerase processivity factor PCNA has also been reported previously" (lane 218).
10. Citation for "which resembles the previously identified ZFP281 recognition motif" (lane 227).
11. Figure S4A, ZFP281 binding has many other sites other than these co-localized with BRCA2. What are these sites? Does it have relation with observed defects in ZFP281 KO cells.
12. Figure S4B and S4C, it seems that these data are not sufficient to draw the conclusion.
13. Figure 6g, h is described in the text but not labeled in the figure.

Reviewer #1 (Remarks to the Author):

This is a priori a nice piece of work in which authors show a new interaction between ZFP281 and BRCA2 with a specific role in the DNA damage response in particular at bivalent chromatin. Authors show a great amount of data to show that a replication defect in cells depleted of ZFP281 pointing out to this defect being mediated by DNA-RNA hybrids, as previously shown for BRCA2-deficient cells. Both proteins would work together to solve the aberrant structure. The study is of interest but it has a number of overinterpretations and results that need to be better supported to claim part of the conclusions.

We appreciate the positive comment from the reviewer. In the meantime, we are truly grateful to the reviewer for pointing out the overinterpretation issue to us. In the revised version, we have either rephrased the text, or provided more evidence to support our conclusion to avoid overinterpretations.

1) The claim of a ZEBRA complex is not supported with the results provided. Authors should either remove the reference to this complex or show its existence using the appropriate biochemistry tools. Authors just make a pull down overexpressing ZFP281 from a plasmid and further mass-spectrometry analysis. However, we need information how resistant to salt concentrations the putative complex is, whether the interaction is observed in samples in which DNA and RNA are removed (there is no indication about this in M&M, and since they conclude that it is formed at bivalent chromatin, it means that they assume that there is DNA in their preps) and a clean gel where we see the purified components with a minimal idea of stoichiometry between the different members. The conclusion that ZFP mediates the assembly of the complex on the bivalent chromatin, is an overinterpretation, because authors show no experiment addressing this question and no conclusion can be made from ChIP studies on the existence of complexes but only on the accumulation of the factors at the same chromatin region, not necessarily as part of a stable complex or even at the same time during the cell cycle.

We have now rephrased this part of text to make it more conservative. We have removed the ZEBRA complex in the text. Instead, we have stated in the revised text that ZFP281 and BRCA2 interact with each other and function together on R-loop.

We also apologize for not making this part of data clear. In fact, we performed FLAG purification experiments under high salt condition in the presence of benzonase to degrade nucleic acids. The extraction buffer contains 420 mM NaCl, and the binding buffer contains 300 mM NaCl. The elution products were directly digested in solution before subject to mass spectrometry. Thus, we were unable to label different factors on the silver staining gel. We also performed the endogenous IP experiments in the high salt buffer and found that the four factors, ZFP281, BRCA2, EMSY and QSER1, can interact with each other. In addition, size-exclusion chromatography analyses indicated that these four factors can be co-fractionated. These were the reasons why we concluded in the original text that ZFP281, BRCA2, EMSY and QSER1 are able to form a complex. We have now made the experimental procedures clear in the revised M&M, and also rephrased this part that ZFP281 and BRCA2 interact with each other.

Since both ZFP281 and BRCA2 play roles on chromatin, we thus performed and analyzed ZFP281 and BRCA2 ChIP-seq, and found 83% of the BRCA2 peaks were also bound by ZFP281. These ZFP281 and BRCA2 co-bound regions can be further divided into two groups according to their chromatin mark, the bivalent regions and the active regions. ZFP281 was required for the binding of BRCA2 to the bivalent regions. Thus, our functional genomics analysis suggest that ZFP281 and BRCA2 might function together in the bivalent regions.

2) The data in Figure 1d is not very convincing as cyclin c levels seem to follow the same pattern of cyclin A2 and E in the gel. The quantification does not seem to fully match the gel. The experiment should be repeated and quantifications should be shown as average signal with error bars. Furthermore, quantitative western blots by fluorescent signals rather than ECL would ensure that the loaded proteins are within the linear range of detection and that the signal is not affected by the different amounts of proteins or the kinetics of the ECL reaction.

We thank the reviewer for pointing out this issue to us. We have now repeated the experiments, measured the western blot signals, and shown average signal with error bars in the revised Figure 1d. We also presented one of the biological replicates in the revised supplementary Figure 2B.

However, to more convincingly claim a cell cycle progression defect, the authors should ideally measure BrdU incorporation profiles by flow cytometry in addition to the cellular quantification of the different S phase patterns in 1e.

According to the reviewer's suggestion, we measured EdU incorporation profiles by flow cytometry in control and ZFP281 knockout ES cells. The percentage of S phase cells was decreased after ZFP281 knockout. The new data has been provided as the new Figure 1c.

3) The data in other western-blot (figure 2d and 3e) should also be properly quantified and reproducible to claim that PCNA levels are reduced upon ZFP281 loss, leading to increased levels of H2AX phosphorylation and to show a slow recovery after APH release and that all these phenotypes are rescued by RNase H overexpression.

We thank the reviewer for this valuable suggestion. We have now first analyzed total, chromatin bound, and soluble fraction PCNA protein level changes after ZFP281 knockout. Our results indicated that total PCNA levels remained unchanged after ZFP281 knockout, and that ZFP281 knockout led to decreased association of PCNA to chromatin. We also measured the western blot signals, and showed average signals with error bars. The new data have been provided as the revised Figure 2d. We also presented one of the biological replicates in the supplementary Figure 3A.

In addition, we have repeated the western blots shown in the original Figure 2d and 3e, measured the western blot signals, and shown average signal with error bars. The new data have been provided as the revised Figure 3e. We also presented one of the biological replicates in the

supplementary Figure 5B.

Additionally, if replication is perturbed in ZFP281 cells, increased fork asymmetry would be expected in DNA combing experiments, as it has been shown in other R-loop accumulating mutants rather than only a slow S phase. This phenotype should be also suppressed by RNase H.

We thank this reviewer for the insightful questions. We agree the conclusion of increased fork asymmetry in R-loop accumulating mutants, but those might not be applicable to aberrant R-loop accumulation at specific genomic regions¹⁻⁵. Given that ZFP281 and BRCA2 co-occupy a subset of bivalent regions, we tried to perform DNA combing assay combined with H3K4me3 and H3K27me3 immuno-staining. Unfortunately, we were unable to obtain images with ideal resolution due to technical limit.

In order to investigate the effect of ZFP281 on DNA replication at specific genomic regions, we performed BrdU-IP in the APH released wild type, ZFP281 KO, and ZFP281 KO+RNaseH overexpression mouse ES cells. Our results suggested that ZFP281 knockout specifically affected the BrdU incorporation after release from APH at the bivalent regions, but not the active regions. The BrdU incorporation defect can be rescued by RNase H1 overexpression. The new data have been provided as the revised supplementary Figure 11C.

In the future, we are still striving to achieve these goals to reach single genomic locus resolution of replication changes with ZFP281 KO, but it will be beyond the scope of this manuscript.

1. Tuduri, S. *et al.* Topoisomerase I suppresses genomic instability by preventing interference between replication and transcription. *Nat. Cell Biol.* **11**, 1315–1324 (2009).
2. Gan, W. *et al.* R-loop-mediated genomic instability is caused by impairment of replication fork progression. *Genes Dev.* **25**, 2041–2056 (2011).
3. Herrera-Moyano, E., Mergui, X., Garcia-Rubio, M. L., Barroso, S. & Aguilera, A. The yeast and human FACT chromatin-reorganizing complexes solve R-loop-mediated transcription-replication conflicts. *Genes Dev.* **28**, 735–748 (2014).
4. Prendergast, L. *et al.* Resolution of R-loops by INO80 promotes DNA replication and maintains cancer cell proliferation and viability. *Nat. Commun.* **11**, 4534 (2020).
5. Tsai, S. *et al.* ARID1A regulates R-loop associated DNA replication stress. *PLOS Genet.* **17**, e1009238 (2021).

4) Figure 2a shows a single line in a single cell. Given the abundant signal detected in both PCNA and ZFP281 immunofluorescences, it is not surprisingly that they overlap to a certain degree. A more robust data should be presented to claim that these two factors co-localize in the cell, such as a PLA analysis with both antibodies.

We thank the reviewer for pointing out this issue to us. We have now presented an image with multiple cells to show the localization of ZFP281 and PCNA (Figure 2a, b). In addition, we performed reciprocal immunoprecipitations and found that ZFP281 and PCNA are able to interact with each other (Figure 2e).

5) From Fig. 2d in page it is not possible to conclude that ZFP281 is required for PCNA loading or stabilization on chromatin during DNA replication, since it may well be that an earlier replication event is delayed so that PCNA loading is affected as a consequence but not because ZFP helps PCNA to be loaded. This needs to be better reasoned and clarified.

We appreciate the reviewer for the thoughtful advice. In this revised version, we demonstrated that ZFP281 and PCNA can interact with each other using the co-IP experiments. In addition, as mentioned above, we analyzed total, chromatin bound, and soluble fraction PCNA protein level changes after ZFP281 knockout. We found that displacement of PCNA from chromatin to the soluble fraction after ZP281 knockout. Thus, it is very likely that ZFP281 is required for PCNA loading on chromatin through directly binding to PCNA. However, as the reviewer suggested, we still cannot rule out the possibility that the levels of PCNA on chromatin were reduced as a consequence of DNA replication defects. To make it more conservation, we have rephrased this part in the revised version as “ZFP281 might be directly required for maintain the proper levels of PCNA on chromatin during DNA replication”.

6) From Figure 2g, the authors state that the BrdU recovery is delayed in ZFP281 KO cells. In contrast, it seems that the peak of maximum BrdU incorporation is at the same time or even earlier in ZFP281 KO cells (green line).

We thank the reviewer for pointing out this to us. We have now rephrased this part in the revised version as “replication recovery from the APH block was affected in ZFP281 KO cells”.

7) Figure 3a should be more than a single cell. Indeed, the second cell partially shown in the WT shows much more S9.6 signal arguing that the chosen cell is not a representative image.

We thank the reviewer for reminding us. We have now presented the images containing more cells in Figure 3a. In addition, we also quantified the R-loop signal intensities in WT (n=48), ZFP281 KO-1 (n=66), and ZFP281 KO-2 (n=53), and performed statistical analyses, as shown in Figure 3b.

8) The authors conclude that ZFP281 recruits BRCA2 and other proteins to form the ZEBRA complex that is important to prevent R-loops at bivalent promoters based on CHIPseq data. However,

-Figure 6f is not cited in the text. Moreover, as a general model, it would be more illustrative to depict the bivalent regions where this phenomenon occurs. Also, drawing certain replisome components and PCNA unloading seems to indicate that the replisome is intact except for PCNA but this is not tested in the paper at all.

We thank the reviewer for reminding us. We have now simplified the Figure 6f model and cited it in the revised text.

-It is unclear the reasoning to validate the ChIPseq data in ZFP281 KO cells (Figure 5f) performing

ChIP-qPCR in shZFP281-depleted cells (Supplementary Figure 7) rather than ZFP281 KO cells or for the DRIP in Figure 5g,h.

We have now changed the order by describing the ChIP-seq/qPCR and DRIP results in ZFP281 knockdown cells, and then confirming the results in ZFP281 KO cells.

-It would be expected that EED KO cells should decrease the signal in DRIP experiments in these bivalent promoters in support of this conclusion.

We thank the reviewer for this suggestion. Indeed, we also thought that PRC2 should be involved in R-loop regulation. Since PRC2 deletion abolished the binding of BRCA2 to the bivalent region, we speculated that PRC2 deletion might cause increased R-loop levels at these regions. However, we and other found that R-loop levels were not affected after deletion of PRC2 subunits (supplementary Figure 11A)¹. It is apparent that deletion of PRC2 leads to changes in chromatin state and transcription, besides loss of BRCA2. In addition, it has been reported that PRC2 can interplay with multiple factors that function in R-loop regulation, such as the helicase ATRX and DDX5. It therefore remains an open question how regulation on R-loop at the PRC2 occupied regions occurs at different layers.

1. Skourti-Stathaki, K. *et al.* R-Loops Enhance Polycomb Repression at a Subset of Developmental Regulator Genes. *Mol Cell* **73**, 930-945 e934 (2019).

9) In general, the quality of the graphical representation is poor. The error bars are barely distinguishable in figure such as 2e,f,g (also 5g,h). Also, the statistics is missing in all these graphs whereas the authors state significant variations throughout the paper. Do the data correspond to the average between independent experiments? How many? This is not indicated at all in the legends or the methods.

We appreciate the reviewer for reminding us. We have performed statistical analyses to the graph data used here in the revised version. Data presented here are the representative results of at least three biological replicates. We have now made these clear in the revised M&M.

10) BRCA2 is a member of the Fanconi Anemia pathway, which works at damage blocking replication. Since this pathway has been shown to resolve also r loops (Schwab et al, Mol Cell 2015; García-Rubio et al. PloS Genet 2015) it would be interesting to discuss the results obtained in this context.

We thank the reviewer for this suggestion. We have now discussed the Fanconi Anemia pathway in the manuscript.

Minor points

-page 12, line 268 should say Figure 5g,h instead of 6g,h

We thank the reviewer for pointing this out. We have corrected this typo.

-Figure S3E should accompany the main Figure 4f to be able to understand it.

According to the reviewer's suggestion, we have moved the schematic illustration of BRCA2 truncations (the original Figure 3E) to the main figure, as the Figure 4f, and changed the original Figure 4f to 4g.

-The Methods section should state where confocal microscopy was used, such as in Figure 2A, where Airyscan imaging is said to have been used in the Results section.

We have now included this information in the M&M section.

-line 227, page 10. A reference should be cited to indicate the previously identified ZFP281 recognition motif.

We have now added the reference.

-The English language should be revised. Some examples of incorrect spelling/grammar include:

-Chromatin is a not countable noun. Thus, 'chromatins' is not appropriate.

-line 81: 'a' is missing before 'specific regulatory mechanism'

-line 90 page 4, 'previously studies' should be 'previous studies'

We thank the reviewer for pointing these mistakes out. We have corrected them all.

Reviewer #2 (Remarks to the Author):

R-loops are unusual but important nucleic acid structures that regulate gene expression and if not tightly regulated can pose serious threats to genome integrity. The authors focused on embryonic stem cells and used cellular and genomic approaches to show that ZFP281 cooperates with BRCA2 in preventing R-loop accumulation to promote DNA replication. They used biochemical purifications that led to the identification of ZFP281 and BRCA2 in a ZEBRA complex. They further show that the enrichment of BRCA2 at G/C-rich promoters requires both ZFP281 and PRC2, which if disrupted leads to perturbations in PCNA loading and replication defects. This is an interesting story and highly relevant to both the DNA repair and R-loop fields, but the authors need to address the following points:

1. The authors suggest that ZFP281 and BRCA2 might both function in preventing R-loop accumulation at the subset of bivalent regions in mouse ES cells but fail to explain why a global increase is there if the function is only at specific genomic loci.

We thank the reviewer for pointing out this to us. Indeed, S9.6 staining in control and ZFP281 knockout indicated that R-loop levels were substantially increased after ZFP281 knockout. It is very possible that R-loops in the bivalent regions might not be the sole targets of ZFP281. We have previously shown that ZFP281 depletion led to de-repression of retrotransposons,

including LINE-1s. Retrotransposons, which make up almost half of the mammalian genome, are R-loop hot spots. Therefore, we cannot rule out the possibility that ZFP281 might be also involved in R-loop regulation in retrotransposons. This could explain why substantial increases in R-loop levels were detected by S9.6 staining after ZFP281 knockout. We have now included this in the discussion section.

2. Can the authors tie the R-loop data to bivalent chromatin?

We appreciate the reviewer for this advice. We have now provided more analyses to tie the R-loop data to the bivalent chromatin according to the reviewer's suggestions below.

E.g. would RNase H1 expression reduce binding of BRCA2 and other ZEBRA components to bivalent domains?

RNaseH1 overexpression led to increased occupancy of ZFP281 and BRCA2 at the tested bivalent regions, but not the active regions, indicating that R-loop and ZEBRA complex might regulate reciprocally at these regions. This set of new data has been provided in the new supplementary Figure 12A, B.

Would RNase H1 overexpression change the expression of PRC-2 bound genes?

We performed RT-qPCR analyses and found the RNaseH1 overexpression slightly de-repressed the expression of these tested PRC2-bound genes. It is in line with a previous study showing that R-loop removal leads to decreased PRC2 recruitment, but not H3K27me3, and increased transcription at the R-loop bivalent genes¹. This set of new data has been provided in the new supplementary Figure 12C.

1. Skourti-Stathaki, K. *et al.* R-Loops Enhance Polycomb Repression at a Subset of Developmental Regulator Genes. *Mol Cell* **73**, 930-945 e934 (2019).

How does ZEBRA resolve R-loops?

It has been reported that the DEAD-box RNA helicase DDX5 is able to interact with BRCA2, and that BRCA2 stimulates the DNA-RNA hybrid-unwinding activity of DDX5 helicase. The interplay between PRC2 and DDX5 has also been reported by different groups. Thus, it is very possible that ZEBRA might resolve R-loops at the bivalent regions through DDX5. We have now included this in the discussion section.

How does the R-loop resolution lead to epigenetic changes at these specific loci?

It has been reported that the epigenetic marks including H3K27me3 remained nearly unchanged after RNase H1 overexpression, possibly due to low turnover rates of epigenetic marks during the short window of the experiments¹. We also performed H3K27me3 and H3K4me3 ChIP after RNase H1 overexpression, and found that, indeed, H3K27me3 was largely unaffected despite

that SUZ12 occupancies were slightly reduced at the tested bivalent regions in the RNase H1 overexpressed cells. This set of new data has been provided in the new supplementary Figure 12D, E.

1. Skourti-Stathaki, K. *et al.* R-Loops Enhance Polycomb Repression at a Subset of Developmental Regulator Genes. *Mol Cell* **73**, 930-945 e934 (2019).

3. It is often not clear if the authors performed biological independent replicates or technical repeats, e.g. no error bars in Fig1b.

We appreciate the reviewer for reminding us. Data presented here are the representative results of at least three biological replicates. We have now made these clear in the revised M&M. We have also added error bars in the revised Figure 1b.

4. I am not convinced that PCNA in chromatin was significantly reduced in the absence of ZFP281 in Fig.2d. This is complicated by variations in loading and the authors need to provide quantification after normalisation and also show PCNA protein levels in the soluble nucleoplasmic fraction which should show a corresponding increase in PCNA.

According to this valuable suggestion, we have now first analyzed total, chromatin bound, and soluble fraction PCNA protein level changes after ZFP281 knockout. Our results indicated that total PCNA levels remained unchanged after ZFP281 knockout, and that ZFP281 knockout led to decreased association of PCNA to chromatin. We also measured the western blot signals, and showed average signals with error bars. The new data have been provided as the revised Figure 2d. We also presented one of the biological replicates in the supplementary Figure 3A.

In addition, we have repeated the western blots shown in the original Figure 2d and 3e, measured the western blot signals, and shown average signal with error bars. The new data have been provided as the revised Figure 3e. We also presented one of the biological replicates in the supplementary Figure 5B.

5. It is not clear in Fig1d that ZFP281 KO exhibits an effect on cyclin E levels. Can the authors provide quantification of three biological repeats and stats?

We appreciate the reviewer for reminding us. We have independently repeated the western blots to analyze the effects of ZFP281 KO on cyclin proteins for three times, measured the western blot signals, and shown average signal with error bars. The new data have been provided as the revised Figure 1d. We also presented one of the biological replicates in the supplementary Figure 2B.

6. The previous comment applies to multiple figures in the manuscript such as Fig2c and Fig2d. Also, I am not sure if the current data supports the conclusion that RNase H1 overexpression could partially rescue the delay of PCNA re-loading to chromatin and abrogate the aberrant increase of γ H2A.X after APH release in ZFP281 KO cells (Fig. 3e). There is no quantification nor statistical

analysis.

According to the reviewer's suggestion, we have now independently repeated these western blots for at least three times, quantified the intensity, and performed statistical analysis.

7. The interaction between ZFP281 and PCNA in Fig. 2a requires validation by co-IP or proximity ligation assays.

We thank the reviewer for the thoughtful advice. In this revised version, we demonstrated that ZFP281 and BRCA2 can interact with each other using the co-IP experiments. Thus, it is very likely that ZFP281 regulates the levels of PCNA on chromatin through directly binding to PCNA.

8. Fig2g - I can't see a delay in recovery. I can see a reduction. Could you comment further on this?

We thank the reviewer for pointing out this to us. We agreed with the reviewer that DNA replication levels after recovery from the APH block at the tested regions were reduced, which are also consistent with the reduced levels of PCNA after treatment. These data suggest that other factors, besides ZFP281, could also contribute to the loading of PCNA during DNA replication. We have now rephrased this part in the revised version as "BrdU IP-qPCR analyses revealed that replication recovery from the APH block was affected in ZFP281 KO cells, as reflected by the reduced levels of nascent DNAs at the ZFP281 bound loci *Gata6*, *Kctd1*, *Lhx4* and *Pou3f2* (Fig. 2g)".

9. The R-loop data in Fig3a need confirmation by an orthogonal method such as DRIP-qPCR

We have performed the DRIP-qPCR after depletion of ZFP281 or BRCA2. The results have been shown in the revised Figure 5g and h, supplementary Figure 11B.

10. Fig3d - Stats and error bars are missing. Should RNase H1 over-expression increase the number of mitotic cells, as the knockout cells are not going through mitosis due to aberrant accumulation of R-loops? Can the authors comment on this discrepancy?

According to the reviewer's suggestion, we added error bars, performed statistical analyses, and found that the number of mitotic cells remained unchanged in ZFP281 KO, and ZFP281 KO + RNase H1 cells (Figure 3d). It is very likely that aberrant accumulation of R-loop after ZFP281 KO leads to early S phase delay, subsequently abnormal S-G2 transition. However, the percentage of M phase cells was not much affected by ZFP281 KO or ZFP281 KO + RNase H1 OE.

11. Fig3e – the images are not convincing. The authors need to provide an RNase H blot and quantification of PCNA and γ H2A.X signal intensity with stats.

We thank the reviewer for this suggestion. We have now provided the RNase H1 blot in WT, ZFP281 KO, ZFP281 KO + RNase H1 cells in the supplementary Figure 5A. We have also repeated the western blot, quantified PCNA and γ H2A.X intensity, and performed the statistical analyses

in the revised Figure 3e.

12. Fig4b – the authors need a negative control: i.e. a blot of a protein that ZEBRA components do not bind to.

We thank the reviewer for this suggestion. We have now repeated the IP experiments with a negative control MEK2, shown in the revised Figure 4b.

13. Fig5f - Can the authors explain what KO-1/WT and KO-2/WT mean? The reduction is also not obvious. A better-quality image and a quantification are needed in this figure.

We apologize that we did not make it clear here. In Figure 5f, we used heat map to show the fold change of BRCA2 occupancy after ZFP281 KO (in ZFP281 KO-1, ZFP281 KO-2 cells). Green color indicates $\log_2 \leq -1.0$, reduced BRCA2 occupancy after ZFP281 KO. Thus, from the figure, we shown that the occupancy of BRCA2 reduced at the bivalent, but not active, regions after ZFP281 KO. We have now labelled the figure to make it clear, and changed “KO-1/WT and KO-2/WT” to “ZFP281 KO-1 vs. WT and ZFP281 KO-2 vs. WT”.

14. Fig5gh - Stats are again missing. Could you also highlight in the figure which primers are 'active' and which ones are 'bivalent'?

According to the reviewer’s suggestion, we have now performed the statistical analyses. We also highlighted the primers of active regions using green color, and bivalent regions using red color.

15. There are multiple typos, and the manuscript will benefit from a careful proofreading, e.g:

- a. Line 90: previously studies
- b. Line 144: for different time

We thank the reviewer for pointing these mistakes out. We have corrected them all.

Reviewer #3 (Remarks to the Author):

The manuscript by Wang et al described a ZFP281 and BRCA2-containing protein complex ZEBRA, providing a potential explanation for the replication defects and increased R-loop formation in ZFP281 KO cells. ZFP281 depletion leads to R-loops accumulation, results in delayed PCNA loading and DNA replication defects, which are also observed in BRCA2 KO cells. Genome-wide occupancy analyses revealed that BRCA2 is highly enriched at G/C-rich promoters and that ZFP281 as well as PRC2 are required for BRCA2 localization at bivalent chromatin. Thus, the ZEBRA complex appears to be required for preventing R-loops and ensures proper progression of DNA replication. The data are of high quality and the conclusion links the replication defects of ZFP281 KO cells to the function of BRCA2.

Major concerns:

1. What are R-loops distributions in ZFP281 and BRCA2 KO cells? What is the percentage of

overlapping? Is R-loops only limited to bivalent chromatin? In this case, PRC2 KO cells should also have increased accumulation of R-loops.

We apologize that we did not make it clear here. R-loops are prevalent in mammalian genomes, not limited to the bivalent regions¹. Here, we found that ZFP281 and BRCA2 co-occupy a subset of bivalent region, and that ZFP281 is required for the recruitment of BRCA2 to these regions. By using DRIP-qPCR, we found that both ZFP281 and BRCA2 are required for repressing R-loop levels at the ZFP281 and BRCA2 co-occupied regions. To what extent the R-loops regulated by ZFP281 and BRCA2 overlap is definitely worth further investigation.

Since PRC2 deletion abolished the binding of BRCA2 to the bivalent region, we also speculated that PRC2 deletion might cause increased R-loop levels at these regions. However, we and other found that R-loop levels were not affected after deletion of PRC2 subunits (supplementary Figure 11A)². It is apparent that deletion of PRC2 leads to changes in chromatin state and transcription, besides loss of BRCA2. In addition, it has been reported that PRC2 can interplay with multiple factors that function in R-loop regulation, such as the helicase ATRX and DDX5. It therefore remains an open question how regulation on R-loop at the PRC2 occupied regions occurs at different layers.

1. Sanz, L.A. *et al.* Prevalent, Dynamic, and Conserved R-Loop Structures Associate with Specific Epigenomic Signatures in Mammals. *Mol Cell* **63**, 167-178 (2016).
2. Skourti-Stathaki, K. *et al.* R-Loops Enhance Polycomb Repression at a Subset of Developmental Regulator Genes. *Mol Cell* **73**, 930-945 e934 (2019).

2. What is the relationship of replication defects in ZFP281 KO cells and the failure of activating of bivalent genes in ZFP281 KO cells upon differentiation?

We apologize that we did not make it clear here. According to literatures, it is very likely that both the replication defects and the failure of bivalent gene activation can arise in ZFP281 KO cells as partial consequences of aberrant R-loop accumulation. It has been well established that R-loops are enriched at the 5' end of those genes with promoter-proximal Pol II pausing, and that R-loop formation promote RNA Pol II pausing, thus regulates transcription¹⁻³. Therefore, the failure of bivalent gene activation in ZFP281 KO cells upon differentiation could be direct consequence of R-loop accumulation, but not DNA replication defects.

1. Sanz, L.A. *et al.* Prevalent, Dynamic, and Conserved R-Loop Structures Associate with Specific Epigenomic Signatures in Mammals. *Mol Cell* **63**, 167-178 (2016).
2. Skourti-Stathaki, K. *et al.* R-loops induce repressive chromatin marks over mammalian gene terminators. *Nature* **516**, 436-439 (2014).
3. Shivji M.K.K. *et al.* BRCA2 regulates transcription elongation by RNA Polymerase II to prevent R-loop accumulation. *Cell Rep* **22**, 1031-1039 (2018).

Specific comments:

1. Figure 1a, which gRNAs are in ZFP281 KO1 and KO2 cells?

We thank the reviewer for reminding us. We have now revised the supplementary Figure 1A to make it clear.

2. Figure 1d, are changes in these cyclins in protein levels or RNA levels? Please provide a quantitative measurement.

According to the reviewer's suggestion, we have now examined the RNA levels of these cyclins after ZFP281 depletion. We found that paralleling its RNA levels, Cyclin E1 protein levels were reduced in the two ZFP281 KO cells, though the RNA levels of Cyclin E2 remained unchanged. The protein, but not RNA, levels of the S phase specific Cyclin A2 were also reduced after ZFP281 KO, while the G1 phase specific Cyclin C was marginally affected. The new figures were provided as Figure 1d and supplementary Figure 2.

3. Figure 1e, please explain: S phase cell is reduced but early S phase cell is increased?

Thanks for the reviewer's question. The proportion of early S phase cells in 281 KO cells was increased, while middle S phase cells reduced, indicating 281 KO delayed the early-to-middle S phase progression, which might affect the whole S phase during cell cycle progression.

4. Lane 131, please explain: however, the EdU positive ratios of entire S phase were comparable between WT and ZFP281 KO cells. This seems contradicted with Figure 1e.

We appreciate the reviewer for pointing this out. Indeed, according to the FACS and EdU staining data in Figure 1c and 1e, the EdU positive cells were slightly reduced after ZFP281 KO. We have now corrected it.

5. How are the R0-R4 related to cell cycle progression?

Thanks for the reviewer's question. Mouse ES cells proliferate with rapid cell cycle kinetics, completing a full cell cycle within 12 hours. The S phase of mouse ES cells require about 8.5 hours¹⁻². DNA polymerase inhibitor aphidicolin (APH) has been widely used to specifically inhibit the elongation of DNA replication, and then block the cycling cells in early S phase^{3,4,5}. In this study, we added 0.3 μ M APH into ES media and culture for 16 hrs, then released the replication block by culturing in APH-free media for 0, 1, 2, 3, 4 hrs respectively. This treatment allowed us to trace the function of ZFP281 during DNA replication, especially the early and middle S phase progression.

1. Fujii-Yamamoto, H. *et al.* Cell cycle and developmental regulations of replication factors in mouse embryonic stem cells. *J Biol Chem* **280**, 12976-12987 (2005)
2. Stead E., *et al.* Pluripotent cell division cycles are driven by ectopic Cdk2, cyclin A/E and E2F activities. *Oncogene* **21**, 8320-8333 (2002).

3. Saintigny Y., *et al.* Characterization of homologous recombination induced by replication inhibition in mammalian cells. *EMBO J* **20**, 3861-3870 (2001)
4. Frum, R.A. *et al.* DNA replication in early S phase pauses near newly activated origins. *Cell Cycle* **7**:1440-1448 (2008)
5. Ahuja, K.A. *et al.* A short G1 phase imposes constitutive replication stress and fork remodelling in mouse embryonic stem cells. *Nat Commun* **7**, 10660 (2016).

6. Supplementary Fig. 2 is not appropriately cited (page 8).

We thank the reviewer for pointing this out. We have now corrected it.

7. Figure 4c, label the molecular size of gel filtration column.

We have now labelled the molecular size in the revised Figure 4c.

8. Figure 4, draw a diagram to indicate the interactions between different proteins (domains).

According to the reviewer's suggestion, we have now drawn a model to indicate the interactions among the four proteins based on the biochemical studies in the manuscript. The model has been provided in the revised in the supplementary Figure 6E.

9. Citation for "co-localization of BRCA2 with the DNA polymerase processivity factor PCNA has also been reported previously" (lane 218).

We have now added the reference.

10. Citation for "which resembles the previously identified ZFP281 recognition motif" (lane 227).

We have now added the reference.

11. Figure S4A, ZFP281 binding has many other sites other than these co-localized with BRCA2. What are these sites? Does it have relation with observed defects in ZFP281 KO cells.

We appreciate the reviewer for this insightful thought. Indeed, ZFP281 binds to many other sites without BRCA2. It is very possible that the binding of ZFP281 to these sites is also associated with the observed defects in ZFP281 KO cells. For example, we have previously found that ZFP281 binds to retrotransposons, and that its depletion led to de-repression of retrotransposons, including LINE-1s. Retrotransposons, which make up almost half of the mammalian genome, are also R-loop hot spots. Therefore, we cannot rule out the possibility that ZFP281 might be also involved in R-loop regulation in retrotransposons. We have now included this in the discussion section.

12. Figure S4B and S4C, it seems that these data are not sufficient to draw the conclusion.

We thank the reviewer for pointing this out. We have now rephrased as “Our ChIP-qPCR analyses demonstrated that EMSY and QSER1 also occupied the randomly selected ZFP281 and BRCA2 co-bound regions in mouse ES cells”.

13. Figure 6g, h is described in the text but not labeled in the figure.

We thank the reviewer for pointing this out. We have now corrected it.

REVIEWER COMMENTS

Reviewer #1 (Remarks to the Author):

Authors have made a good work adding controls requested to support their conclusions or reducing/removing part of conclusions (like the ZEBRA reference). The manuscript has thus certainly improved providing interesting data of cross-reactions between ZFP281, BRCA2 and PRC2 on R loop formation as a way to explain the role of PRC1 in genome dynamics. However, it is still unclear how ZFP281 protects against R loops. The increasing number of proteins being published lately with an impact on R loops, suggests that apart from a nice correlation between the sites of action of BRCA2 and ZFP281, the ms does not provide a mechanism to understand how ZFP281 prevents or resolves R loops. This is not solved by just focusing the study on one of the factors that appear in the proteomic analysis. In addition, since BRCA2 is a DSB repair factor and the binding of ZNF281 to ssDNA may be related to replication deficiency, it is possible that the action of BRCA2 is required for repair of damage accumulated after ZFP281 inactivation, the accumulation of the R loop being a consequence. Indeed some results do not fit with their model, such as that BRCA2 is irrelevant for ZFP281 occupancy or that RNH1 overexpression elevated the occupancy of ZFP201 and BRC2 to the bivalent regions. If authors consider that BRCA2 could represent a bridge with ZFP281 to bring DDX5 to resolve the R loop, this should be demonstrated, but I am afraid that without a biochemical analysis to provide a biological meaning to the interaction between ZFP281 and BRCA2, we will not have a definitive answer. Thus, this is a nice report with an amazing amount of work using standard approaches with yet another factor, but without a rational mechanism to explain rationally the data and a model that is not supported by the data.

Reviewer #2 (Remarks to the Author):

The authors have included additional experiments and performed repeats, quantifications and missing statistical analyses of original data, which led to notable improvement of the manuscript. One point remains to be addressed: the data showing direct interaction between ZFP281 and PCNA in Figure 2e are not entirely convincing. An orthogonal assay is required such as PLA. Does the interaction increase or decrease after DNA damage or R-loop induction?

Reviewer #3 (Remarks to the Author):

The authors performed a large number of experiments and addressed most of my concerns. I only have several minor questions.

1. For Figure 4c, the description of the gel filtration is not accurate. 2 MDa is the void volume of the Super 6 column. The complex cannot be bigger than 2 MDa.
2. Figure 1b, the two cell lines used in this study show dramatic differences in proliferation, yet exhibit similar levels of cyclin changes. Can the author comment on this?
3. Figure 1c, the changes in the S phase are very minor (68% to 61%). how can this explain the cell growth defects?

Reviewer #1 (Remarks to the Author):

Authors have made a good work adding controls requested to support their conclusions or reducing/removing part of conclusions (like the ZEBRA reference). The manuscript has thus certainly improved providing interesting data of cross-reactions between ZFP281, BRCA2 and PRC2 on R loop formation as a way to explain the role of PRC1 in genome dynamics. However, it is still unclear how ZFP281 protects against R loops. The increasing number of proteins being published lately with an impact on R loops, suggests that apart from a nice correlation between the sites of action of BRCA2 and ZFP281, the ms does not provide a mechanism to understand how ZFP281 prevents or resolves R loops. This is not solved by just focusing the study on one of the factors that appear in the proteomic analysis. In addition, since BRCA2 is a DSB repair factor and the binding of ZNF281 to ssDNA may be related to replication deficiency, it is possible that the action of BRCA2 is required for repair of damage accumulated after ZFP281 inactivation, the accumulation of the R loop being a consequence. Indeed some results do not fit with their model, such as that BRCA2 is irrelevant for ZFP281 occupancy or that RNH1 overexpression elevated the occupancy of ZFP201 and BRC2 to the bivalent regions. If authors consider that BRCA2 could represent a bridge with ZFP281 to bring DDX5 to resolve the R loop, this should be demonstrated, but I am afraid that without a biochemical analysis to provide a biological meaning to the interaction between ZFP281 and BRCA2, we will not have a definitive answer. Thus, this is a nice report with an amazing amount of work using standard approaches with yet another factor, but without a rational mechanism to explain rationally the data and a model that is not supported by the data.

We truly and deeply appreciate the positive comments and insightful questions from the reviewer. We here summarized the reviewer's questions, and responded point-by-point.

1) Please provide a mechanism to understand how ZFP281 prevents or resolves R loops.

In the revised manuscript, we have demonstrated that ZFP281 was also able to interact with DDX5, besides BRCA2, by immunoprecipitation assays (Fig. 5h). In addition, we found that the interaction between BRCA2 and DDX5 was substantially reduced after ZFP281 KO (Fig. 5i). Thus, one of the possible explanations is that ZFP281 was required for the interaction between BRCA2 and DDX5, and thus preventing R-loop accumulation.

2) Is it possible that the action of BRCA2 is required for repair of damage accumulated after ZFP281 inactivation, the accumulation of the R loop being a consequence?

We apologize that we did not make the logic clear here. We showed in the manuscript that ZFP281 KO led to defective PCNA chromatin loading and early DNA replication, and unscheduled R-loop accumulation. In order to investigate whether R-loop accumulation could be one of the possible reasons of replication defects after ZFP281 KO, we overexpressed RNase H1 in ZFP281 KO cells to remove R-loops, and found that R-loop removal was able to rescue the PCNA chromatin loading defects and reduce γ H2A.X level in ZFP281 KO cells (Fig. 3e). We have now discussed it in the revised version.

3) Some results do not fit with their model, such as that BRCA2 is irrelevant for ZFP281 occupancy or that RNH1 overexpression elevated the occupancy of ZFP281 and BRCA2 to

the bivalent regions.

We thank the reviewer for reminding us about the model. We have now revised the model to show that ZFP281 is required for BRCA2 occupancy at the bivalent regions, while BRCA2 is irrelevant for ZFP281 occupancy; we have now also included DDX5 in the model. Most importantly, we added an additional panel indicating that RNase H1 overexpression in ZFP281 KO cells is able to rescue the DNA replication defects caused by ZFP281 KO.

Reviewer #2 (Remarks to the Author):

The authors have included additional experiments and performed repeats, quantifications and missing statistical analyses of original data, which led to notable improvement of the manuscript. One point remains to be addressed: the data showing direct interaction between ZFP281 and PCNA in Figure 2e are not entirely convincing. An orthogonal assay is required such as PLA. Does the interaction increase or decrease after DNA damage or R-loop induction?

We totally agreed with the reviewer, and thus immediately ordered the PLA kit from Sigma after receiving the reviewer's comments. However, due to the Covid-19 pandemic, the delivery date has been postponed again and again, and the kit is still intercepted by the customs. Thus, we asked Prof. Zhou Zhongjun in the University of Hong Kong (HKU) for help.

Prof Zhou and colleagues in HKU independently performed the PLA assay to probe the direct interaction between ZFP281 and PCNA in mouse ESCs. The ZFP281-PCNA PLA fluorescent puncta were formed and evident in mouse ES cells. APH treatment reduced the number of the ZFP281-PCNA PLA fluorescent puncta (Fig. 2f). Thus, our analyses validated the interaction between ZFP281 and PCNA, and also showed that the interaction between ZFP281 and PCNA was compromised once DNA replication was blocked.

Reviewer #3 (Remarks to the Author):

The authors performed a large number of experiments and addressed most of my concerns. I only have several minor questions.

1. For Figure 4c, the description of the gel filtration is not accurate. 2 MDa is the void volume of the Super 6 column. The complex cannot be bigger than 2 MDa.

We apologize that we did not make the column information clear. We used the high resolution Superose 6 increase 3.2/300 column for the gel filtration analysis, which can characterize proteins with molecular weights between 5 KDa and 5 MDa. We have now included the detailed column information in the revised manuscript.

2. Figure 1b, the two cell lines used in this study show dramatic differences in proliferation, yet exhibit similar levels of cyclin changes. Can the author comment on this?

The S phase specific Cyclin E and Cyclin A2 were reduced to similar levels in both KO cells, which is consistent with flow cytometry assays indicating similar reduction of S phase cells after ZFP281 KO. Compared to WT cells, both of the ZFP281 KO cell lines showed reduction in cell numbers over culture. Indeed, the effect of ZFP281 KO-2 was greater. However, since Cyclin E and Cyclin A2 are S phase specific, we might not be able to use the changes of these two cyclins to fully explain

the reduced numbers of ZFP281 KO cells over culture. It is possible that other cyclins/factors might also contribute to the different degrees of reduction in the numbers of the two KO cells.

3. Figure 1c, the changes in the S phase are very minor (68% to 61%). how can this explain the cell growth defects?

We thank the reviewer for the question. Flow cytometry analyses indeed showed a slight reduction in S phase cell fraction after ZFP281 KO. To further investigate whether ZFP281 is required for proper S phase progression during cell cycle, we identified early, middle and late S phase cells according to the spatiotemporal pattern of EdU incorporation. Compared with WT ES cells, ZFP281 KO cells exhibited a significant increase in the percentage of early S phase cells. In contrast, the proportion of middle S phase cells was decreased in ZFP281 KO cells (Fig. 1e). Therefore, ZFP281 KO led to more severe early S phase defect, which could at least partially explain the growth defects in ZFP281 KO cells.

REVIEWERS' COMMENTS

Reviewer #1 (Remarks to the Author):

The authors have improved the manuscript by adding mainly new data affecting BRCA2-DDX5 interaction and doing some corrections. They have changed the model to attend the criticisms, which sounds a bit strange how easily they makes changes to a model. IN any case, I do not understand why in the revised manuscript, the changes are not clearly marked in the text to allow reviewers to focus with more details on the changes, considering that this is the third time we have to go thru this ms. For instance, authors indicate that they have changes in the Discussion. But they did not mark them. Thus, it is difficult to understand that in Discussion no mention at all is made to DDX5, when it seems to be an important piece in their model. INdeed they claim to have apossible explanation in the rebuttal, but I do not see it in the Discussion.

Revise refrences. For exmplae, Ref 25 uo?

Reviewer #2 (Remarks to the Author):

The authors have addressed all my concerns. It is, however, still not entirely clear how BRCA2 acts as a bridge with ZPF281 to bring DDX5 to resolve the R loop, so I suggest this part should be toned down in the discussion.

Reviewer #1 (Remarks to the Author):

The authors have improved the manuscript by adding mainly new data affecting BRCA2-DDX5 interaction and doing some corrections. They have changed the model to attend the criticisms, which sounds a bit strange how easily they makes changes to a model. IN any case, I do not understand why in the revised manuscript, the changes are not clearly marked in the text to allow reviewers to focus with more details on the changes, considering that this is the third time we have to go thru this ms. For instance, authors indicate that they have changes in the Discussion. But they did not mark them. Thus, it is difficult to understand that in Discussion no mention at all is made to DDX5, when it seems to be an important piece in their model. INdeed they claim to have a possible explanation in the rebuttal, but I do not see it in the Discussion.

Revise references. For exmplae, Ref 25 uo?

We truly and deeply appreciate the positive comment from the reviewer. We here summarized the reviewer's questions, and responded point-by-point.

1) About the model

We thank the reviewer for reminding us about the model in last reviewer's comments. That's why we modified the model to summarize and better present our results. Compare the previous and current versions of model, the conclusion of this manuscript remains same, but readability and intelligibility of the cartoon model might be improved.

Previous version of model

Current version of model

2) About DDX5 in the model

We apologized sincerely for the confusion. We previously discussed the possibility of ZFP281 and BRCA2 could resolve R-loop through DDX5. In the last revision, we confirmed the interaction of DDX5, ZFP281 and BRCA2, and demonstrated that ZFP281 is required for interaction between DDX5 and BRCA2. So we moved this part from the discussion to the result section, highlighted the change in the result section, and added DDX5 into the model.

3) About the reference

We are very grateful to the reviewer for pointing this to us. We have now corrected Ref 25.

Reviewer #2 (Remarks to the Author):

The authors have addressed all my concerns. It is, however, still not entirely clear how BRCA2 acts as a bridge with ZPF281 to bring DDX5 to resolve the R loop, so I suggest this part should be toned down in the discussion

We truly and deeply appreciate the positive comment and the insightful suggestion from the reviewer. We have now toned down DDX5 in the discussion.